# CloudRoots: Integration of advanced instrumental techniques and process modelling of sub-hourly and sub-kilometre land-atmosphere interactions

Jordi Vilà-Guerau de Arellano[1,2], Patrizia Ney[3], Oscar Hartogensis[1], Hugo de Boer[4], Kevin van Diepen[1], Dzhaner Emin[5], Geiske de Groot[1], Anne Klosterhalfen[6], Matthias Langensiepen[7], Maria Matveeva[5], Gabriela Miranda-García[1], Arnold F. Moene[1], Uwe Rascher[5], Thomas Röckmann[8], Getachew Adnew[8], Nicolas Brüggemann[3], Youri Rothfuss[3], Alexander Graf[3]

[1]Meteorology and Air Quality Section, Wageningen University Research, 6708 PB Wageningen, the Netherlands
[2]Atmospheric Chemistry Department, Max Planck Institute for Chemistry, 55128 Mainz, Germany
[3]Institute of Bio- and Geosciences, IBG-3: Agrosphere, Forschungszentrum Jülich GmbH, 52425 Jülich, Germany
[4]Department of Environmental Sciences, Faculty of Geosciences, Utrecht University, 3508 TA Utrecht, the Netherlands
[5]Institute of Bio- and Geosciences, IBG-2: Plant Sciences Forschungszentrum Jülich GmbH, 52425 Jülich, Germany
[6]Department of Forest Ecology & Management, Swedish University of Agricultural Sciences, 90183 Umeå, Sweden
[7]Faculty of Agriculture, University of Bonn, 53115 Bonn, Germany
[8]Institute of Marine and Atmospheric Research Utrecht University, 3584 CC Utrecht, the Netherlands

*Correspondence to*: Jordi Vilà-Guerau de Arellano (jordi.vila@wur.nl)

**Abstract.** The CloudRoots field experiment was designed to obtain a comprehensive observational data set that includes soil, plant and atmospheric variables to investigate the interaction between a heterogeneous land surface and its overlying atmospheric boundary layer at the sub-hourly and sub–kilometre scale. Our findings demonstrate the need to include measurements at leaf level to better understand the relations between stomatal aperture and evapotranspiration (ET) during the growing season at the diurnal scale. Based on these observations, we obtain accurate parameters for the mechanistic representation of photosynthesis and stomatal aperture. Once the new parameters are implemented, the model reproduces the stomatal leaf conductance and the leaf-level photosynthesis satisfactorily. At the canopy scale, we find a consistent diurnal pattern on the contributions of plant transpiration and soil evaporation using different measurement techniques. From the high frequency and vertical resolution of state variables and carbon dioxide ($CO_2$) measurements, we infer a profile of the $CO_2$ assimilation in the canopy with non-linear variations with height. Observations taken with a laser scintillometer allow us to quantify the non-steadiness of the surface turbulent fluxes during the rapid changes driven by perturbation of photosynthetically active radiation by cloud flecks. More specifically, we find two-minute delays between the cloud radiation perturbation and ET. To study the relevance of advection and surface heterogeneity for the land-atmosphere interaction, we employ a coupled surface-atmospheric conceptual model that integrates the surface and upper-air observations made at different scales from leaf to the landscape. At the landscape scale, we calculate a composite sensible heat flux, by weighting measured fluxes with two different land-use categories, which is consistent with the diurnal evolution of the boundary-layer depth. Using sun-induced fluorescence measurements, we also quantify the spatial variability of ET and find large variations at the sub-kilometre scale around the CloudRoots site. Our study shows that

throughout the entire growing season, the wide variations in stomatal opening and photosynthesis lead to large diurnal variations of plant transpiration at the leaf, plant, canopy and landscape scales. Integrating different advanced instrumental techniques with modelling also enable us to determine variations of ET that depend on the scale where the measurement were taken and on the plant growing stage.

5 ## 1. Introduction

Evapotranspiration (ET), the net exchange of water vapour between the land and the atmosphere, remains an elusive process to be measured, quantified and represented in models because of it depends on the interaction of multiple processes that act in a wide range of scales (Katul et al., 2012). ET is a key variable in the exchange of heat, moisture and carbon dioxide at the surface and it strongly depends on how radiation and energy are partitioned into latent and sensible heat (Moene and Dam, 10 2014; Monson and Baldocchi, 2014). The amounts of direct and diffuse radiation reaching the leaves depend on the transfer of radiation that is strongly perturbed by clouds and aerosols, and on its subsequent penetration into the canopy. Triggered by ambient light conditions, the stomatal responses coupled to the surface and boundary-layer dynamics is the main driver that regulates how the net available radiative energy is partitioned between the turbulent sensible and latent heat fluxes (van Heerwaarden and Teuling, 2014). However, due to the highly non-stationary nature of atmospheric radiation (van Kesteren, 15 et al., 2013b) and turbulent nature of the meteorological fluctuations, we still lack a fundamental understanding of the two-way feedback between stomatal control and cloud radiation perturbations across scales and land/atmosphere conditions (Katul et al., 2012; Sikma et al., 2018).

The bi-directional link between surface processes and boundary layer clouds as described above is what we refer to as the CloudRoots concept, where boundary-layer dynamics and clouds are rooted in, or coupled to, the surface and vice-versa 20 (Vilà-Guerau de Arellano et al, 2014). The degree of coupling depends on soil, plant and weather conditions characterized by the diurnal variability of wind, temperature and specific humidity (Sikma et al., 2018). To fully comprehend this system requires inclusion of all necessary parameters at the required spatial scales, from the size of the stomata (10 - 100μm) to the depth of the boundary layer and cloud top (~3 km), temporal scales from seconds to daily and seasonal cycles and across disciplines bringing together experts from ecophysiology to turbulence. This can only be obtained by integrating 25 experimental and modelling efforts. Here we describe and show first results of the CloudRoots field experiment aimed at obtaining new understanding about the interaction between the soil, vegetation and the clear/cloudy boundary layers at these sub-hourly and sub-kilometre scales, *i.e.* on spatiotemporal scales smaller than the characteristic grid resolution scales of the weather (typical resolution ranging from 1 to 10 km) and climate (typical resolution ranging from 20 to 100 km) models. In that respect, CloudRoots field campaign continues the tradition of experiments that connect land surface properties with 30 boundary-layer dynamics, but now using advanced instrumental techniques and modelling the coupling between the essential processes. Two examples of such previous campaigns are the First ISLSCP Field Experiment (FIFE) (Hall et al., 1989) and the Boreal Ecosystem-Atmosphere Study (BOREAS) (Sellers et al., 1995).

Thanks to their high-quality routine measurement program (Franz et al., 2018; Rebmann et al., 2018), ICOS sites lend themselves as anchors for additional experiments. Here, we describe the CloudRoots campaign near the agricultural site Selhausen (ICOS site DE-RuS) and the Jülich Observatory for Cloud Evolution – Core Facility (JOYCE, http://joyce.cloud) in Germany during spring 2018 (Löhnert et al., 2015). In order to quantify all the necessary scales of interest - leaf, canopy and landscape-, we complemented the existing radiation, flux and soil measurements of the ICOS site by scintillometry, microlysimeters, sap-flow and leaf-level flux measurements, quasi-instantaneous vertical profiles and spectroscopic measurements of vegetation indices and sun-induced fluorescence (SIF). Scintillometers provided minute-scale turbulent fluxes enabling us to connect stomatal responses to the energy, moisture and $CO_2$ fluxes at this timescale. Microlysimeters, soil flux chambers, sap-flow, leaf-level chambers and canopy-resolving profile all have the ability to distinguish vegetation from soil $CO_2$ and water vapour ($H_2O$) fluxes in contrast to the eddy-covariance technique that provided net fluxes from the two sources combined. The remote sensing measurements of boundary-layer dynamic evolution and cloud properties made at JOYCE provided evidence on diurnal variations of the boundary-layer depth, the role of entrainment and cloud diurnal variability. A key aspect of the research strategy of CloudRoots is the integration of all these measurements in a land-atmosphere conceptual model CLASS (Vilà-Guerau de Arellano et al., 2015). This model has been specially developed to support the interpretation of measurements at the sub-hourly scales (Vilà-Guerau de Arellano et al., 2019).

To this end, we study the following five facets of the diurnal interactions between the land and the atmosphere: (i) observational validation at leaf level of the mechanistic model representation of the stomatal aperture and photosynthesis, (ii) the diurnal variability of $H_2O$-$CO_2$ flux partition due to the soil and plant contributions at the canopy level, (iii) the no-steadiness of these fluxes due to the influence of clouds, (iv) the spatial heterogeneity of ET inferred from the SIF measurements and (v) the integration of the observations in the conceptual model CLASS to quantify the influence of of land-surface heterogeneity and advection. We finally obtain a daily estimation of ET and discussed differences with respect to the observational or modelling techniques.

The paper is organized as follows. In Section 2 we give a detailed overview of the field experiment with special emphasis on the instrumentation used that serve the overall goals of our CloudRoots concept. The results Section 3 is organized along the five topics outlined above. First, at leaf level, we validate a photosynthesis-conductance mechanistic model that is commonly used in large-eddy simulations (Pedruzo-Bagazgoitia et al., 2017; Sikma et al., 2018) and the global numerical model prediction system ECMWF-IFS (Boussetta et al., 2013). This allows us to assess the need to revisit currently used constants in the mechanistic model representing photosynthesis. This part is completed by comparing leaf transpiration rate with tiller-level measurements of sap flow at different stages of the growing season. Second, and in order to scale up to the canopy level, we analyse the soil and plant partitioning of the net ET and net ecosystem exchange (NEE) based on the inversion of observed high-resolution vertical concentration profiles (Warland and Thurtell, 2000; Santos et al., 2011). Third, in analysing the impact of clouds on ET, we measure the potential effectiveness of diffuse radiation in enhancing ET and NEE (Kanniah et al., 2012). Extending previous work by van Kesteren et al. (2013b), we quantify the time-lag between fluctuations in incoming shortwave radiation and ET in the field. These real-world measurements are an essential addition to

time-lag of plant responses to radiation changes studied in laboratory experiments (Vico et al., 2011). Fourth, we infer the spatial variability of ET around the CloudRoots site using SIF remote-sensing observations. Fifth, all these observations are then integrated in several numerical experiments made by CLASS with special emphasize on the treatment and role of how to include surface heterogeneity and heat/moisture advection to improve the interpretation of the observations. Finally, in the discussion Section 4 we bring together and discuss all CloudRoots methodologies by comparing their daily ET estimates. Conclusions are given in Section 5.

## 2. Description of the Cloud Roots field experiment and modelling effort

### 2.1 Site description

The CloudRoots field campaign was carried out at the Terrestrial Environmental Observatory (TERENO) Selhausen, which is located in the southern part of the Lower Rhine Embayment in Western Germany (50°52'09"N, 6°27'01"E, 104.5 m altitude) in a region largely dominated by agriculture (Fig. 1). In 2011, the site was equipped with micrometeorological measurement devices for long-term monitoring of energy and carbon exchange. Since 2015, the station has been extended in accordance with ICOS standards for Level 1 sites (ICOS site code DE-RuS) (Ney et al., 2019). For this campaign, a further IRGASON eddy-covariance (EC) system with an open path gas analyser (see Sect. 3.4) was placed on the test field and used for additional flux measurements presented here.

The test field covered 9.8 ha and was surrounded by other croplands (Ney and Graf, 2018). As Fig. 1 shows, these cultivated areas comprise mainly winter wheat, winter barley, sugar beet, rapeseed, maize, potatoes and peas, whereby the various field sizes and locations of crops has led to small-scale heterogeneity in the vegetation cover. An agricultural road, mainly used by farm machinery, passes by the northern edge of the field. The next inhabited settlement is located 500 m to the west (Fig. 1a). There are two lignite open-cast mines in the wider surrounding of the study site, located 6 km northeast (extension of 4400 ha with a maximum depth of 470 m b. g. l.) and 6 km west (extension of 1400 ha with a maximum depth of 200 m b. g. l.). In general, the land surface at the study site is flat and has a slope less than 4°. A loess layer with a thickness of about 1 m covers Quaternary sediments, which were mainly built-up from fluvial deposits of the Rur river system. The overlying soil is an Orthic Luvisol according to the USDA classification (IUSS Working Group WRB, 2006), whose texture is silt loam with a mixture of 14% clay, 73% silt and 13% sand (Schmidt et al., 2012).

The local climate is classified as temperate maritime with an annual mean air temperature of 10.3°C and an annual mean precipitation of 718 mm (reference period 1981-2010, data taken from the DWD climate station of the Forschungszentrum Jülich 5.3 km distant from the test site). The observation period from beginning of May until end of June 2018 was characterized by a 2.9°C higher mean air temperature (17.5°C) and 46% less precipitation in comparison to the long-term average. Fig. 1b shows the heterogeneity quantified by the sensible heat fluxes measured at the CloudRoots site and a bare soil field nearby. In consequence, and as shown by Fig 1c, in CloudRoots we aim to integrate horizontal and vertical scales in the analysis of ET and its relation to boundary-layer dynamics.

The field campaign covered the main growing phases (booting, heading and maturity stage) of winter wheat. During the observation period, we did three intensive observation periods (IOP). During these IOPs the following complementary instruments and measurements were added: microlysimeters, leaf-level measurements, SIF measurements on canopy and regional scale, as well as vertical profiles of state variables and $CO_2$ within and above the canopy were performed. Fig. 2 shows a timeline of the deployment of the campaign-specific measurement setup (see Sect. 3.4) that includes the IOPs on 7[th] May (IOP 1), 15[th] (IOP 2) and 28[th] June 2018 (IOP 3). The main meteorological and biometric conditions are summarized in Table 1. The test field was cultivated with a crop rotation cycle typical of the region (Ney et al., 2019). The rotation prior to the observation period was beet/potatoes/winter wheat (catch-crop) and sugar beet. Residues of the harvest of sugar beet were left on the site and ploughed in before the cultivation cycle started with the sowing of winter wheat (*Triticum aestivum* L.; variety Premio) in October 2017. The field was fertilised with mineral nitrogen (N) once in March, April and May 2018 (81.6, 39.2 and 50 kg N ha[-1], respectively). The wheat was harvested on 17 July 2018 with a yield of 92 dt ha[-1]. A detailed overview of the field management practices before, during and after the campaign is given in the Appendix (Table A1).

## 2.2 Weather and crop description during the IOPs

The weather situation during all three IOPs was mainly characterized by an anticyclonic pressure pattern over Central Europe (IOP 1 and IOP 2), extending up to Northern Europe during IOP 3, which led to high 2 m-temperatures up to 24 to 26°C during IOP 1 and IOP 2, and 28°C during IOP 3 (Table 1). Cloudiness and temperature-inversion heights at the top of the atmospheric boundary layer were different. While weak subsidence motions during IOP 1 led to a slightly rising temperature-inversion layer between 1200 to 2000 m abobe ground level (a. g. l.) with clear conditions during the whole period (mean daytime global radiation $S\downarrow$ of 514 W m$^{-2}$), a weak cold front passed the measuring site from the northwest in the early morning of IOP 2 (mean daytime $S\downarrow$ of 311 W m$^{-2}$). Diurnal heating caused the replacement of a layer of stratocumulus at a height of 1800 m a. g. l., in the morning, followed by the appearance of scattered towering cumulus clouds. Light showers occurred only in the vicinity of the site. During IOP 3, a few shallow cumulus and cirrus clouds appeared, despite the existence of a small upper-air low which passed the area around the edge of a larger cut-off, although it was located above South-Eastern Europe. The mixed boundary layer was topped at a height of around 1700 m a. g. l.

The persistent high-pressure weather conditions resulted in a drought during the entire observation period. Ongoing dryness led to a reduction in the soil water content at 20 cm depth (Table 1) from 27 vol.% during IOP 1 to 15 vol.% at IOP 3. Maturity occurred 14 days earlier than in previous years. The leaf area index (LAI) ranged from 4.5 (green growing stage) m$^2$ m$^{-2}$ in IOP 1 to 5.5 m$^2$m$^{-2}$ IOP 2 (green/yellow ripening stage). No changes in LAI were observed between IOP 2 and IOP 3 (yellow/senescence stage).

## 2.3 Instrument description

Table 2 summarizes all the variables measured and modelled during CloudRoots, together with specific nomenclature and information on units and scales.

### 2.3.1 Microlysimeters

For direct measurements of soil evaporation ($E_{lys}$), four microlysimeters were installed at a number of locations around the EC-station (one in each cardinal direction) at the beginning of every observation period. In order to obtain an undisturbed soil monolith for each microlysimeter, an SDR-35 polyvinyl chloride (PVC) collar with an inner diameter of 0.2 m, a wall thickness of 0.005 m, and a depth of 0.11 m was pushed carefully into the ground. Afterwards the collar including the soil column was retrieved, its outside was cleaned, and the bottom of each lysimeter was sealed with an acrylic glass disc, which prevented percolation and capillary rise from or into the microlysimeter. The microlysimeters were then weighed initially and returned to their original positions. We made sure that the lysimeters were levelled with the soil surface, their walls fully surrounded by soil, and that the crop was affected and destroyed as little as possible, so that the general conditions and characteristics of the field site could still be maintained (e.g., regarding heat flux, shading). All four microlysimeters were subsequently collected, cleaned, weighed and distributed again every sixty or ninety minutes. A scale with a precision of 0.1 g (equivalent to 0.00318 mm evaporation) was used. The scale was enclosed in a box to avoid wind effects during the measurements. Finally, the measured weight differences were converted to W m$^{-2}$ by means of the lysimeters surface area, the time periods between weighing and the latent heat of vaporization (Quade et al., 2019).

### 2.3.2 Soil $CO_2$ flux chambers

Soil respiration ($R_s$) was observed with an automated soil $CO_2$ gas flux system (Li-8100, Li-Cor Inc. Biosciences, Lincoln, Nebraska, USA), connected to four long-term soil flux chambers. The chambers were installed close to the EC-station (one in each cardinal direction) on top of PVC soil collars with a diameter of 0.2 m and a total height of 0.07 m, from which 0.05 m was inserted into the soil. Each chamber was closed at thirty-minute intervals for 90 seconds during flux measurements, while $CO_2$, water vapour concentrations and chamber headspace temperature were recorded at a sampling rate of 1 Hz. The $CO_2$ concentration was standardized to dry air and a constant temperature, to eliminate effects of changes in air density and water vapour dilution during closure time. $R_s$ was subsequently calculated by adjusting a linear regression fit to the final 60 seconds of the measurement before reopening.

### 2.3.3 Leaf-level measurements

Leaf gas exchange was measured using a Li-Cor LI-6400XT portable photosynthesis system with a 6400-02B LED light source. Leaf-level measurements included instantaneous stomatal conductance to water vapour ($g_{sw}$) and photosynthesis ($A_{leaf}$), maximum light-saturated photosynthesis ($A_{max}$), $CO_2$-response curves and light-response curves. Measurements of $g_{sw}$ and $A_{leaf}$ were performed during the three IOPs, starting at sunrise and ending when measurements of $g_{sw}$ indicated that stomata had nearly closed ($g_{sw} < 0.05$ mol m$^{-2}$ s$^{-1}$). For measurements of $g_{sw}$ and $A_{leaf}$, tillers were picked randomly in the field and immediately mounted in the leaf chamber for measurements. Initial tests showed no difference in $g_{sw}$ between excised and attached tillers. Settings of leaf chamber photosynthetically active radiation (PAR) and $CO_2$ followed the diurnal

variability measured in the field. For comparison with other observations, measurements of $g_{sw}$ and $A_{leaf}$ were binned and averaged at thirty-minute intervals. Maximum light-saturated photosynthetic capacity ($A_{max}$) was measured during the three IOPs as well as on 8[th] May between 10:00 and 12:00 UTC. For measurements of $A_{max}$ the light intensity (PAR) was set to 1500 µmol m$^{-2}$ s$^{-1}$ and the leaf was equilibrated under a reference $CO_2$ concentration of 450 µmol $CO_2$ mol$^{-1}$air. $CO_2$ response curves were measured during IOP 1 and IOP 3 prescribing $CO_2$ concentrations in the following order: 450, 50, 100, 150, 250, 350, 450, 600, 800, 1200 µmol$CO_2$ mol$^{-1}$air. All $CO_2$-response curves were measured using a light intensity (PAR) of 1500 µmol m$^{-2}$ s$^{-1}$. Light-response curves were measured on IOP 1 only and used a reference $CO_2$ concentration of 450 µmol$CO_2$ mol$^{-1}$air. PAR values were changed in the following order: 0, 25, 50, 100, 200, 400, 800, 1200 1500 µmol m$^{-2}$ s$^{-1}$. The stomatal conductance to water vapour ($g_{sw}$ [mol m$^{-2}$ s$^{-1}$]) of the A-PAR curves in between 0-200 µmol m$^{-2}$ s$^{-1}$ for the three repetitive experiments within the PAR range were (average and standard (deviation in brackets): 0.49 (0.13), 0.12(0.02) and 0.34(0.06). Leaves were allowed to equilibrate to leaf chamber conditions in terms of gas exchange (approximately one to two minutes), but not in terms of stomatal aperture. For all measurements, leaf chamber temperature was set between 20°C and 25°C. Relative humidity in the leaf chamber was set between 60% and 75%. Measurements of $A_{max}$, $CO_2$-response curves and light-response curves were performed on attached tillers.

### 2.3.4 Sap-flow

Sap-flow in wheat tillers was measured with the heat-balance method (Sakuratani 1981; Baker and van Bavel, 1987). Twenty-four tillers were selected at random, diameters measured with an electronic calliper and SGA3-type sap-flow sensors installed at the lowest possible internodes following the procedure recommended by the manufacturer (Dynamax, 2007). Sensors were connected with electrically shielded wired to AM 16/32 multiplexers controlled and scanned by CR1000 data loggers (Campbell Scientific, Logan, Utah, USA). Energy supply to the stem heaters was carefully regulated to the highest permissible level in order to obtain a strong heat signal. We employed the dual voltage regulators (Dynamax AVRDC) which were parts of wired measurement, control and extension units assembled and tested by the heat-balance sensor manufacturer (Flow32 1K A and B models, Dynamax Inc., Houston, Texas USA) Data were processed according to the calculation procedure of Dynamax (2007) with adaptations to wheat (Langensiepen et al. 2014) to obtain reliable data on the convective stem heat flow generated by sap flow. Here we take the evolution of the tiller densities from 480 tillers m$^{-2}$ (IOP 1 and IOP 2) to 370 tillers m$^{-2}$ (IOP 3) into account.

### 2.3.5 Profiling-elevator

Vertical profiles $H_2O$ and $CO_2$ expressed as mole fractions $\chi H_2O$ and $\chi CO_2$ (mole of substance per mole of moist air), temperature ($T_{air,p}$) and wind speed ($u_p$) from the soil surface to the surface layer above the crop canopy were measured with a portable elevator system. The elevator moved continuously up and down the measuring sensors attached to an extension arm over a total profile height of 2 m. A sampling tube connected to a differential gas analyser (LI-7000, Li-Cor Inc. Biosciences, Lincoln, Nebraska, USA) collected $\chi H_2O$ and $\chi CO_2$ at a frequency of 20 Hz. $T_{air,p}$ and $u_p$ were measured at the

same frequency by a ventilated fine wire thermocouple (FW3, Campbell Scientific, Logan, Utah, USA)and a hotwire anemometer (8455-075-1, TSI, Shoreview, Minnesota, USA). All measurements were duplicated as a continuous fixed-height measurement at the top of the profile. During the data post-processing, the temporal and vertical resolution of the mean profiles was set to a time-averaging block of thirty minutes with a vertical resolution of 0.025 m. Time delays in each variable with respect to the position caused by response times of the sensors, electronic delays and the tube transport of the gas samples were adjusted by a hysteresis minimization algorithm. Detailed information on the profile measurement setup and the processing the data profile is given in Ney and Graf (2018). The measured concentration profiles were then used to determine the vertical source profiles of $H_2O$ and $CO_2$, with the aim of providing an independent, non-invasive partitioning between aboveground net primary production (NPP) and $R_s$ or evaporation (E) and transpiration (Tr). To estimate source profiles and flux partitioning we used an analytical dispersion Lagrangian technique introduced by Warland and Thurtell (2000) and further developed by Santos et al. (2011). Other than in the abovementioned literature, a simple optimization method (Nelder and Mead, 1965) was used to fit four parameters: soil source, canopy source and shape parameters p and q of a beta distribution which describes the vertical source distribution within the canopy.

### 2.3.6 Scintillometer

The receiver of a displaced-beam laser scintillometer, hereafter referred to as DBLS (SLS-20, Scintec, Rottenburg, Germany), was placed 9 m south-east from the EC station (Fig. 1). The scintillometer measurements height was 1.95 m a. g. l.. The path length towards the instrument transmitter was 86.8 m. It was pointed along North-West to South-East. The DBLS measures the scintillation intensity of two displaced laser-beams (wavelength of 670nm and separation distance of ~2.7mm). The structure parameter of temperature ($C_T{}^2$) and dissipation rate of turbulent kinetic energy (ε) are determined from the log-variance of one beam and log-covariance between the beams,. The general equation that links the scintillometer measurements to fluxes is given by:

$$F_x = \rho K_x \left(u_*, \frac{z}{L}\right) z^{\frac{1}{3}} \sqrt{C_{x^2}^2} \tag{1}$$

where $F_x$ is defined as the turbulent flux of the transported variable x, $C_x{}^2$, is the structure function parameter of x, and $K_x$ represents the turbulent exchange coefficient that links $F_x$ to $C_x{}^2$. $K_x$ is a function of the friction velocity, $u_*$, and the Obukhov length, L. Finally ρ is the air density and $z$ the measurement height above the surface. For the sensible heat flux, **H**, x represents temperature (T) and appropriate constants need to be added to convert Eq. (1) to energy fluxes H, $u_*$ and L are solved iteratively as a function of the DBLS measured $C_T{}^2$ and ε (Thiermann, 1992; Hartogensis et al., 2002). The Monin-Obukhov Similarity Theory (MOST) functions that define $K_x$ were taken from Kooijmans and Hartogensis (2015). For our purpose, however, the exact shape of the MOST functions is of minor importance as we are primarily interested in the dynamic, temporal behaviour of the fluxes rather than an accurate description of their quantitative values. We are aware that

advective contributions can lead to the violation of MOST. However, advection was not influencing our measurements for two reasons. First, the scintillometer transmitter and receiver are far enough from the edges of the CloudRoots field given the height of the sensor (1.95 m), the wind speed and direction during the IOPs, and the stability conditions. All of these make that footprints are small enough to fit within the field. Typical footprint length (90% footprint contribution) for the 3 IOPs

yields: IOP 1 (85 m), IOP 2 (30 m) and IO P3 (75 m). Second, the scintillometer has a path weighting function that is maximum in the middle of the path and near-zero at the transmitter and receiver positions, i.e. the major contribution occurs at the farthest point of the field edge.

The added value of DBLS fluxes over the traditional EC method is that they converge to statistically stable flux estimates at much shorter flux averaging times of one minute or less, while the EC technique typically requires flux averaging times of

ten to thirty-minutes (Hartogensis et al 2002; van Kesteren et al., 2013b). The essence behind this is that the flux estimate is based on structure parameters which are defined in the inertial range of the turbulent spectrum. As such the flux estimates rely on a limited range of the turbulent scales that contribute to the flux rather than all as is the case with the EC method.

We also adopted the combination technique introduced by van Kesteren et al. (2013a, 2013b) to obtain fluxes of $H_2O$ and $CO_2$ at these short time scales. This technique combines structure parameters of $H_2O$ and $CO_2$ which are obtained from $H_2O$

and $CO_2$ time-series from an Infra-Red Gas Analyser (IRGASON system; see Sect. 2.3.7) with an exchange coefficient defined by the DBLS fluxes to finally calculate flux estimates of $H_2O$ and $CO_2$. In other words, with $u_*$ and $L$ solved with the DBLS, Eq. (1) can be evaluated using structure parameters of trace gases $x$, where in this case x represents the specific density, $q_x$, of $H_2O$ or $CO_2$.

### 2.3.7 Eddy-covariance and ancillary micrometeorological measurements

A continuously running EC system was operated in the middle of the field (Fig. 1), comprising a three-dimensional sonic anemometer (Model CSAT-3, Campbell Scientific, Inc., Logan, Utah, USA) and an open path infrared gas analyser (Model LI-7500, Li-Cor, Inc., Biosciences, Lincoln, Nebraska, USA). The sensors height was 2.34 m a. g. l. Raw data were sampled in 20 Hz mode and fluxes and averages were calculated as thirty-minutes block averages using the TK3.11 software package developed at the University of Bayreuth, including corrections and quality control as given in Mauder et al. (2013). Missing

values in the calculated turbulent fluxes were filled with the marginal distribution sampling (MDS) method following Reichstein et al., (2005) which is implemented in the REddyProc software package (Wutzler et al., 2018). The station also included measurements of all components of the radiation budget (NR01, Hukseflux, Delft, the Netherlands), PAR (LI-190R, Li-Cor Inc. Biosciences, Lincoln, Nebraska, USA and BF5, Delta-T Devices, Cambridge UK), air temperature ($T_{air}$) and humidity (HMP45C, Vaisala Inc., Helsinki, Finland) at 2.4 m, and precipitation (Thies Clima type tipping bucket,

distributed by Ecotech, Bonn, Germany) at 1.0 m a. g. l.. Radiation measurements were taken at 2.5 m. Soil heat flux, temperature and moisture were measured next to the station (3 x HFP01SC at 3 and 8 cm, Hukseflux, the Netherlands, 3 x TCAV, Campbell Scientific, Logan, USA, 1 cm, 5 cm and 2 to 65 cm layer average, 2 x CS616, Campbell Scientific, Logan, USA, 2 to 6 cm layer average), but also at five points distributed across the field using the wireless SoilNet sensor system

(Bogena et al., 2010). One SoilNet point was placed next to the station, while the other four were placed next to the soil $CO_2$ efflux chambers described above. Each SoilNet point comprised a single soil heat flux measurement at 5 cm (HFP01SC, see above) and combined temperature and soil water content measurements in depths of 1, 5, 10, 20, 50 and 100 cm (SMT100, Truebner GmbH, Neustadt, Germany).

A second mobile EC station with instruments heights of 1.93 m a. g. l. was deployed in the immediate vicinity of the continuously monitoring station during the measurement campaign. The system comprised an IRGASON EC system (SN1185 Irgason EC150, Campbell Scientific, Inc., Logan, Utah, USA; PTB101B pressure sensor, Vaisala Inc., Helsinki, Finland) with an additional LI-7500 sensor (same manufacturer). Here, fluxes were processed with the LiCor EddyPro v6.2.2 software. Radiation (CM11 for global and CG2 for long wave radiation, Kipp & Zonen B.V., Delft, Netherlands),

ground heat flux (4 x HFP01SC at 5 cm depth, Hukseflux, the Netherlands) and temperatures at depths of 2 cm (4 x) and 8 cm (2 x) were also measured at this station.

### 2.3.8 Canopy-level measurements of reflectance and sun-induced fluorescence (SIF): FloxBox

A field spectroscopy system was used (FLOX, JB Hyperspectral Devices UG, Düsseldorf, Germany) for canopy-level measurements of reflectance and SIF. FLOX is constructed for high temporal frequency acquisition of continuous top-of-

15 canopy optical properties with a focus on sun-induced chlorophyll fluorescence. The system is equipped with two spectrometers: an Ocean Optics FLAME S, covering the full range of Visible and Near-Infrared (VIS-NIR) and an Ocean Optics QEPro, with a high spectral resolution (Full Width at Half Maximum – FWHM - of 0.3 nm) in the 650-800 nm range of the fluorescence emission. The optical input of each spectrometer is split between two fibre optic cables, that lead to a cosine receptor that measures solar irradiance and a bare fibre bundle that measures the target-reflected radiance.

Spectrometers are housed in a Peltier thermally regulated box to keep the internal temperature lower than 25 °C in order to reduce dark current drift. The signal is automatically optimized for each channel at the beginning of each measurement cycle and two associated dark spectra are collected as well. Metadata such as spectrometer temperature, detector temperature and humidity, Global Positioning System (GPS) coordinates and time are also simultaneously stored in the secure digital memory of the system. More detailed information about the system can be found in Wohlfahrt (2018) and in Campbell

(2019).

### 2.3.9 Regional level measurements of reflectance and sun-induced fluorescence (SIF): HyPlant

An airborne high performance imaging spectrometer (HyPlant) was used for regional level measurements of the same quantities. Several flight lines over the 15 km x 15 km study site with 1-3 m pixel resolution. HyPlant is a hyperspectral imaging system for airborne and ground-based use, developed as a cooperative effort between Forschungszentrum Jülich

(Germany) and the company SPECIM (Oulu, Finland). It consists of two sensor heads, named DUAL and FLUO. The DUAL module is a line-imaging push-broom hyperspectral sensor, which provides contiguous spectral information from 370 nm to 2500 nm in a single device that utilizes a standard objective lens with 3 nm spectral resolution in the VIS/NIR spectral

range and 10-nm spectral resolution in the SWIR spectral range. The FLUO module measures the vegetation fluorescence signal with a separate push-broom sensor which produces data at high spectral resolution (0.25 nm) in the spectral window between 670 and 780 nm. The position and altitude sensor (GPS/INS sensor) provides, synchronously with the image data, aircraft position and altitude data for image rectification and geo-referencing. Both imagers are mounted in a single platform

with the mechanical capability to align the field of view (FOV). A more detailed description of the sensor is given in Rascher et al. (2015).

Sun-induced fluorescence ($F_{687}$ and $F_{760}$) was retrieved in the two oxygen absorption bands according to the iFLD method. Surface reflectance and vegetation indices were calculated after an atmospheric correction using the MODTRAN software package was applied. The atmospheric correction was performed using the MODTRAN software package (for an overview

of the data processing of HyPlant see Siegmann et al. 2019). For the reasons of easier comparison of SIF values with other methods of this paper, the commonly used SIF units ($mW\ m^{-2}\ sr^{-1}\ nm^{-1}$) were replaced by $nmol\ m^{-2}\ sr^{-1}\ s^{-1}$ using conversion factors 6.35 for $F_{760}$ and 5.74 for $F_{687}$, respectively.

### 2.3.10 Boundary-layer and cloud remote sensing measurements

JOYCE remote sensing facility (Löhnert et al., 2015) (located at a distance of 5 km from the test site) provided continuous

information about boundary-layer and cloud characteristics. Specifically, microwave and LIDAR measurements were used to compare the CLASS model results (see next section) with the inferred boundary-layer depth. This comparison was completed by vertical profiles measured by the routine radio soundings at Essen (station ID EDZE/10410 at a distance of 75 km).

### 2.4 Modelling from leaf to landscape scales: CLASS

The Chemistry Land-surface Atmosphere Soil Slab (CLASS, https://classmodel.github.io/) is a model that couples the soil-vegetation-atmospheric processes and is used to interpret the observations and analyse the interaction of scales (Vilà-Guerau de Arellano, et al., 2015). It contains a leaf-level representation of photosynthesis and stomatal aperture (leaf resistance). By upscaling this leaf resistance to the canopy level (surface canopy resistance), it connects with the soil processes and boundary-layer diurnal dynamics. In 2.4.1 and 2.4.2 we will subsequently discuss the two main modules of CLASS that we

will target in this paper, i.e. the leaf level photosynthesis module and the mixed layer module.

### 2.4.1 Modelling leaf-level photosynthesis

Leaf-level photosynthesis was modelled using the representation of photosynthetic biochemistry, as included in CLASS (Vilà-Guerau de Arellano et al., 2015), which was originally developed by Goudriaan (1986) and further adapted to meteorological applications by Jacobs and de Bruin (1997). As this model describes the relationship between stomatal

conductance ($g_s$) and photosynthesis (A), it is usually referred to as the A-$g_s$ sub-model. In short, plant transpiration and $CO_2$ assimilation as part of the surface energy balance model are represented by a two-big leaves model, one for sunlit leaves and

one for shaded leaves (Jacobs and de Bruin, 1997; Pedruzo-Bagazgoitia et al., 2017). The exchange at the leaf surface depends on the gradient of atmospheric $CO_2$ and an internal leaf $CO_2$ concentration which depends on the water-vapour deficit, and leaf conductance. The $CO_2$ exchange is upscaled to the canopy level by integrating over the leaf area index (LAI).

Available field measurements were used for improving the model settings at the leaf level. The parameters representing the initial value of the light-use efficiency ($\alpha_0$) and the temperature-normalized maximum leaf-level photosynthesis rate ($A_{m,max298}$) were fitted using light-response curves (Fig. 5), and $CO_2$-response curves (Fig. 3b) collected on 8[th] May 2018 (one day after IOP 1), respectively. Table 3 summarizes the optimized values used in the A-$g_s$ (sub)model to simulate the leaf-level photosynthesis. The A-PAR curves contain only the lower light intensity values (0-200 µmol m$^{-2}$ s$^{-1}$) for which the

light response is near-linear and not limited by $CO_2$ diffusion into the leaf. As leaf-level measurements of $A_{max}$ indicated a decline in photosynthetic capacity in the course of the growing season (Fig. 5c), we performed additional measurements of $A_{m,max298}$ to represent the observed seasonal decline for IOP 2 and IOP 3. The impact on these optimized values are shown and discussed in Section 3.5.

## 2.4.2 Modelling the diurnal variability of landscape surface fluxes and boundary-layer dynamics

The fundamental assumption of the mixed-layer model is that under convective conditions the atmospheric boundary layer (ABL) dynamics lead to profiles of the meteorological state variables that are uniform (well-mixed) with height. As a result, these state variables are governed by horizontally averaged 0-dimensional slab equations: one equation for the evolution through time of the slab variable and another for the difference between the residual layer (in the morning transition) and the free tropospheric values and the slab value, *i.e.* the jump at the interface between residual layer and ABL. The ABL

dynamics are governed by the mixed-layer equations of potential temperature (heat), specific humidity (moisture), $CO_2$ and two horizontal wind momentum components. In addition, there is an equation that governs the boundary-layer growth which depends on the buoyancy flux at the surface and the jump in the virtual potential temperature at the interface between the atmospheric boundary layer and the free troposphere.

A key feature of the model is its representation of the sub-daily variability of the land-atmosphere interactions (van

Heerwaarden et al., 2010; Vilà-Guerau de Arellano et al., 2015). The net ecosystem exchange is calculated as a result of the assimilation of $CO_2$ by plants and the $CO_2$ soil efflux. We calculate the assimilation rate from photosynthesis and the stomatal aperture measurements at leaf level (see previous section), up-scaled to canopy level (Ronda et al., 2001). This model depends on the diurnal variability of PAR, temperature ($T_{air}$ and $T_{air,p}$) and the water-vapour deficit (VPD). The two-big leaves approach is used (sunlit and shaded) to take the different contributions of direct and diffuse radiation into account

(Pedruzo-Bagazgoitia et al., 2017). The soil efflux is calculated as a function of the soil temperature and moisture. Other relevant physical processes include a radiation transfer model, the Penman-Monteith equation included in the surface energy balance, and the possibility of adding large-scale forcings such as vertical subsidence motions and large-scale advection of momentum, heat, moisture and $CO_2$. Within the context of CloudRoots, it is important to mention that the model assumes a

horizontal homogeneous surface. While the experimental field itself is quite homogeneous, it is surrounded by other land-use types at a spatial scale that will affect the boundary layer. In that respect, and in setting the initial and boundary conditions for the numerical case, we assume that the boundary layer dynamic is governed by a sensible heat flux that is an aggregate of all the fields shown in Fig. 1b.

## 3 Results: Integrating spatiotemporal scales from leaf to boundary layer

This section is structured following the five facets of the diurnal interactions between the land and the atmosphere outlined in the introduction.

### 3.1 Leaf-level exchange of $H_2O$ and $CO_2$: observations and modelling

We combine leaf-level and sap flow measurements of tiller assimilation and transpiration with leaf-level assimilation modelled by CLASS, A-$g_s$ representation, to study their variation during the growing season and the impact of unsteady PAR due to the presence of clouds.

### 3.1.1 Stomatal conductance and sap flow

Our leaf-level measurements revealed clear diurnal patterns in $g_{sw}$ during all the IOPs (Fig. 3). The observed daily maximum $g_{sw}$ decreased over the growing season. This daily maximum $g_{sw}$ occurred at an earlier time during each IOP. Specifically, the thirty-minute average daily maximum $g_{sw}$ declined from 0.84 mol m$^{-2}$ s$^{-1}$ (around 10 UTC, 12 local time LT) during IOP 1 and 0.83 mol m$^{-2}$ s$^{-1}$ (around 10 UTC) during IOP 2 to 0.30 mol m$^{-2}$ s$^{-1}$ (in between 5:30 and 6:30 UTC) during IOP 3. The weather during IOP 2 was characterized by large cumulus clouds passing over the field site, which were made visible in the large fluctuations in PAR (Fig. 3b, 11 and 12). The cloud-related changes in light intensity induced consistent stomatal opening-closing responses during IOP 2. The relatively low $g_{sw}$ observed during IOP 3 probably reflects the continuing drought that characterized the 2018 growing season in combination with the relatively high VPD and high temperatures. Sap flow measurements were performed during IOP 2 and IOP 3 (Figs. 3b and 3c), and one earlier non-IOP day (7th June) (Fig. 4). Measurements of sap flow revealed clear diurnal patterns for all measurement days and consistent responses to cloud-induced changes in light intensity during IOP 2 (Fig. 3b). These responses were comparable to the observed responses in $g_{sw}$ during IOP 2. Interestingly, the notable decline in leaf-level $g_{sw}$ between IOP 2 and IOP 3 was neither reflected in the measurements of sap flow, nor ET measurements with the eddy-covariance. For IOP 3, the ET measured by the eddy-covariance had still maximum values of 300 W m$^{-2}$. Thereafter, the decrease on ET started one week after (5th July) with values lower than 100 W m$^{-2}$. This discrepancy could partly be explained by increases in VPD and wind speed between IOP 2 and IOP 3. The more probable causes are senescence effects on physiological control of transpiration and the physical reactions to heat of the wheat tillers which were noticeably wilting between IOP 2 and IOP 3. This observation has not been

so far reported in the literature. Further studies of the relationships between senescence and simultaneously occurring changes in the heat-physical properties of wheat tillers are needed to explain this phenomenon.

### 3.1.2 Observed versus modelled leaf-level photosynthesis

One of the main aims in CloudRoots is to improve the mechanistic modelling of photosynthesis and stomatal aperture. To this end, we calibrate the constants of the A-$g_s$ model using systematic in-situ field observations. Fig. 5 shows the dependencies of leaf-level photosynthesis of $A_{leaf}$ on PAR (Fig. 5a) and the leaf-internal $CO_2$ concentration (Fig. 5b), and the long-term decline in maximum light-saturated photosynthesis (Fig. 5c). Our observations indicate the need to calibrate the model depending on the functional type of the plant, in particular the dependence of $A_{leaf}$ on PAR, during the field campaign. Table 2 summarises the new constant values used in the A-$g_s$ model adjusted to the winter wheat crop conditions. Fig. 6 shows a comparison of the model results of $A_{leaf}$ using the new constants and the measurements of $A_{leaf}$ and NPP together with the diurnal variation in PAR and VPD during the three IOPs. Our measurements and model results of $A_{leaf}$ showed clear diurnal patterns during each IOP, and a consistent decline over the three IOPs. The decline in $A_{leaf}$ was comparable to the decline in $A_{max}$ (Fig. 5c) and probably reflects a combination of seasonal decay in photosynthetic capacity and increasing stomatal limitations owing to persistent drought, especially during IOP 3. The magnitude of the seasonal decline in $A_{leaf}$ was comparable to the seasonal decline in NPP derived from EC data. Cloud-induced changes in PAR during IOP 2 also induced changes in $A_{leaf}$. The A-$g_s$ model reproduced the diurnal patterns in $A_{leaf}$ during each IOP as well as the cloud-induced changes in $A_{leaf}$ during IOP 2. The agreement is very satisfactory during IOP 1 characterized by cloudless conditions and the maturity of winter wheat. The model underestimated $A_{leaf}$ during IOP 3, which was a result of the strong stomatal limitations that influenced the measurement of $A_{max}$ on which the model parameterisation from IOP 3 was based. The model furthermore overestimates the decline in $A_{leaf}$ between 14:00 and 19:00 UTC, which probably reflects a misrepresentation of the temperature and VPD sensitivity of *Triticum aestivum*.

### 3.2 Canopy-level partitioning of the net H₂O and CO₂ fluxes between soil and plant processes

Moving from leaf to canopy scale, we analyse the detailed profiles of micrometeorology and carbon dioxide collected using the elevator and infer vertical assimilation profiles as well as the diurnal variability in the surface contributions to ET and NEE.

### 3.2.1 Concentration profiles of H₂O and CO₂, temperature and wind speed

Fig. 7 shows selected thirty-minute mean profiles of $\chi H_2O$ and $\chi CO_2$, temperature and wind speed versus height (z) above ground level during IOP 1 and IOP 2. Over the diurnal cycle, $\chi CO_2$ concentrations fell between 08:00 and 13:00 UTC from 370 to 360 μmol mol$^{-1}$ in the mid canopy during IOP 1 but stagnated slightly below 370 μmol mol$^{-1}$ during IOP 2. This seasonal reduction in $CO_2$ uptake was also observed in measured $A_{leaf}$, i.e. see the decrease of the maximum values in Fig. 6.

The lowest values were observed during local noon, simultaneously with the highest PAR values (Fig. 5b). $\chi CO_2$ minima were located in the upper third of the canopy during IOP 1 and during the middle third during IOP 2. The highest $\chi CO_2$ values were found near the soil surface due to soil respiration, lower light intensity caused by shadowing and a low amount of photosynthetic organs in the stems. Maximum $\chi CO_2$ concentrations were measured in the morning and evening hours and peaked at about 475 and 420 $\mu mol\ mol^{-1}$ during IOP 1 and IOP 2, respectively. The photosynthetic $CO_2$ uptake by plants is highly related to plant transpiration. Consequently, $\chi H_2O$ in the canopy space was higher than in the air above the canopy. The highest values were found directly above the soil surface and were caused by evaporation and within the canopy due to plant transpiration.

The highest temperatures appeared near the canopy top (Fig. 7d, 6e, 7j and 7l). In the late morning of IOP 2, the temperature reached a distinct maximum just below the canopy top (Fig. 7j). This phenomenon has been reported in previous studies (Ney and Graf, 2018) and is caused by the changing solar incidence angle. A low angle of incidence in the morning and afternoon limited the heating to an area just below the canopy surface. Previous studies have shown that the presence of such a pronounced temperature maximum has the potential to increase thermal stability within the canopy and thus inhibit the vertical turbulent exchange of sensible heat (Gryning et al., 2001; Ney and Graf, 2018; Sikma et al. 2020). It can be assumed that the sensible heat flux within the dense plant stand was largely determined by the entire canopy. In other words, during the day, mixing near the soil surface was impeded by stable temperature stratification while in the evening, cooling expanded upwards from the soil surface (Fig. 7f). In general, the processes described above were more pronounced during IOP 2 with its greater canopy height than with the lower canopy during IOP 1. The vertical wind profile showed consistently low wind speeds within the dense canopy ($< 0.5\ m\ s^{-1}$). Above the canopy layer, the wind speed increased in a log-like profile up to a maximum of 2 m s$^{-1}$.

### 3.2.2 Profiles of gross primary production

The detailed profile observations presented in the previous section enable us to calculate height resolved estimates of gross primary production A. Using the 30 min-averages of the vertical profiles for temperature, moisture, and $CO_2$ in the canopy, A is determined using the A-g$_s$ model (Jacobs et al., 1997; Ronda et al., 2001). A (mg m$^{-2}$ s$^{-1}$) is calculated as follows:

$$A = LAD\ (A_m(h) + R_d(h)) \left[1 - exp\left(\frac{-\alpha PAR(h)}{A_m(h) + R_d(h)}\right)\right] \tag{2}$$

where LAD (m$_{leaf}^2$ m$^{-3}$) is the leaf area density, $A_m(h)$ is the $CO_2$ primary productivity (mg m$_{leaf}^2$ s$^{-1}$) as a function of height h, $R_d(h)$ (mg m$_{leaf}^2$ s$^{-1}$) the $CO_2$ dark respiration as a function of h, $\alpha$ (mg J$^{-1}$) is the light use efficiency, and PAR(h) (W m$_{leaf}^{-2}$) is the amount of available photosynthetically active radiation within the canopy. Solar zenith angle related variation in PAR intrusion and differences between atmospheric and skin values for temperature, moisture, and $CO_2$ are neglected. Fig. 8a shows the winter wheat LAD applied in the calculation.

Fig. 8b shows that the entire canopy contributes to the photosynthetic activity, but with maximum A at $h/h_c = 0.7$ ($h_c$: canopy height). This is primarily caused by the extinction of PAR within the canopy and reduced leaf density distribution close to the ground (Fig. 8a). Maximum diurnal productivity is found at around $h/h_c = 0.7$, with the diurnal maximum at 12:00 UTC. Integration over the canopy shows minor discrepancies with respect to the bulk A-$g_s$ model calculation, as the profile data allows for a more precise evaluation of photosynthetic activity. The profile measurements combined with Eq. (2) therefore allows for an improved modelling of the photosynthetic $CO_2$ uptake of vegetation depending on height and the understanding of mechanisms. More accurate estimates of $CO_2$ gross primary production still require improved knowledge of plant canopy micrometeorology (Drewry et al., 2014) .

### 3.2.3 Profile based partitioning of $H_2O$ and $CO_2$

Fig. 9 shows the measured fluxes of latent heat, NEE and soil respiration, as well as their partitioning based on the inversion of vertical high-resolution concentration profiles into the evaporation/transpiration and $R_s$/NPP components. In this section, positive values indicate a flux from the surface/plants into the atmosphere and vice versa. During IOP 1, measured latent heat flux ($L_vE$, hereafter referred to as $ET_{ec}$) showed a typical daily pattern under clear sky conditions (Fig. 9a) with maximum $ET_{ec}$ at noon (345 W m$^{-2}$). Evaporation E of both methods displayed comparable values in the morning and evening but differed at midday. In the morning, the evaporation estimated using the profile measurements and method ($E_p$) and the lysimeter observations ($E_{lysi}$) both consistently suggested low E/ET fractions with E below 10 W m$^{-2}$. Towards noon, $E_p$ increased to 25 and $E_{lysi}$ to 60 W m$^{-2}$, and in the afternoon $E_{lysi}$ reached a maximum of $101 \pm 41$ W m$^{-2}$ (no $E_p$ available). Estimated $Tr_p$ increased to about 290 Wm$^{-2}$ at 11:00 UTC, this being the highest diurnal proportion of ET. Lower $Tr_p$ levels around 12:00 UTC are probably due to a sub-optimal performance of the profile-based partitioning at this particular time. For example, none of the available inversion methods, including the algorithm by Santos et al. (2011) used here, includes the effect of local thermal stability varying with height. Fig. 7 demonstrates that thermal stability increased from the canopy top towards the ground around noon of IOP 1 (Fig. 7e), which may have contributed to the large increase of humidity towards the surface (Fig. 7b) due to the lack of mixing.

Variations in $CO_2$ fluxes NEE, NPP and $R_s$ during IOP 1 are shown in Fig.9b. $NEE_{ec}$ followed a typical diurnal cycle, with strong negative fluxes during the day and slightly positive values (carbon source) during transition times. The highest NEE was observed before noon (-25 µmol m$^{-2}$ s$^{-1}$). $NPP_p$ followed the graph of $NEE_{ec}$, with higher values (-26 µmol m$^{-2}$ s$^{-1}$) in the morning hours than during the afternoon under comparable PAR values. This behaviour coincides with the photosynthesis rate observed at leaf level in Fig. 6a and provides further evidence that carbon uptake by plants was limited due to stomatal occlusion caused by the increase in VPD (Fig. 6a) and/or $T_{air}$ in the afternoon. Profile-based $R_{s,p}$ ranged between 0.5 to 6 µmol m$^{-2}$ s$^{-1}$ with higher values around noon. Compared to measured $R_{s,ch}$, $R_{s,p}$ lay within the standard deviations of $R_{s,ch}$, though $R_{s,p}$ was significantly lower during the morning and evening hours.

### 3.3 Effects of clouds on surface turbulent fluxes

#### 3.3.1 Cloud-induced diffuse fertilisation effect on evapotranspiration

One of the main aims of CloudRoots was to obtain observational evidence of the effects of clouds on the $CO_2$ assimilation and ET. Fig. 10 shows the net primary production (NPP) (left) and $L_vE$ (right), both measured using the eddy-covariance, observed under a wide range of clear and cloudy skies as a function of PAR and compared to Q*at the top of the canopy (van Diepen and Moene, 2019). We analyse a two-week period of observations, between 7[th] May and 20[th] May 2018. The effect of the different direct and diffuse radiation due to cloud perturbations is distinguishable with an enhancement of NPP under clear conditions whereas $L_vE$ is reduced. Clouds affect plant photosynthesis by increasing the fraction of diffuse solar radiation that arrives at the top of the canopy (Kanniah et al., 2012). With a larger contribution of diffuse solar radiation, and within the canopy, the radiation spreads more equally over all leaves and thereby increasing the light-use efficiency of a canopy (Farquhar & Roderick, 2003). At a constant level of radiation at the top of the canopy, the increased light-use efficiency results in enhanced canopy photosynthesis which is known as the diffuse fertilisation effect (Roderick et al., 2001). This phenomenon is especially noticeable for canopies with a high LAI (Knohl & Baldocchi, 2008; Dengel & Grace, 2010). In CloudRoots, and due to the high values of LAI (values in between 4.5 to 5.5), we expect situations in which diffuse fertilisation occurs, but here the question is how it influences $L_vE$. Previous large-eddy simulation modelling studies by Pedruzo-Bagazgoitia et al. (2017) have shown that under conditions dominated by clouds with a small optical depth, *i.e.* thin clouds, $L_vE$ is enhanced with respect to its clear-sky values at the same radiation level.

We find that the observed $L_vE$ is higher, rather than lower, during clear conditions (less diffuse light) than under more diffused cloudy conditions. At constant $Q^*$, the median of $L_vE$ is always higher under clear skies than for cloudy skies. The diffuse fraction plays a minor role and the decrease on $L_vE$ under cloudy conditions is mainly due to the reduction in the incoming shortwave radiation. Our observations indicate that $L_vE$ is driven by the partitioning of direct and diffuse radiation, but also other effects such as diurnal variations of temperature and the link to VPD may partially compensate for the different distribution of direct and diffuse radiation caused by clouds. The higher VPD values during the day partly offset the more optimal PAR conditions and therefore cause a closing of the stomatal that leads to decreases in $L_vE$. For both clear and cloudy skies, the shaded area below the median represents conditions before 11:30 UTC and the shaded area above the median represents conditions after 11:30 UTC, i.e. implying a hysteresis loop (Zhang et al., 2014). This spread in $L_vE$ at a constant level of $Q^*$ is caused by a difference in VPD between morning (before 1130 UTC) and afternoon (after 1130 UTC). This is because on a clear day the VPD raised rapidly due to its non-linear dependence on temperature relative to a cloudy day. In a typical clear day at CloudRoots, the value of 200 W m$^{-2}$ for $Q^*$ is crossed twice: once in the morning and once in the afternoon. When 200 W m$^{-2}$ is crossed in the morning, the VPD is around 1000 Pa and reaches a value of 2000 Pa in the afternoon. On the other hand, on a cloudy day with similar values of around 200 W m$^{-2}$ the VPD remains almost constant through the entire day and with a value of 1000 Pa at 11:30 UTC.

The influence of VPD on $L_vE$ also has the effect that the diurnal cycles of $Q^*$ and $L_vE$ are out of phase due to its dependence on leaf temperature. $Q^*$ is primarily a function of incoming shortwave radiation and VPD of air temperature at the leaf surface. As a result, $Q^*$ and VPD peak at different times of the day. $Q^*$ peaks at maximum incoming shortwave radiation (local noon is at 11:30 UTC), and near-surface VPD times when air temperature peaks, which is around the time at which $Q^*$

$= 0$ (17:00 UTC). The diurnal cycle of the sun implies there is a short period around 11:30 UTC when $Q^*$ does not change. On the contrary, air temperature increases almost linearly around 11:30 UTC due to the approximately constant $Q^*$, as does VPD. Therefore, peak values for $L_vE$ are found between the moments of maximum $Q^*$ and of maximum VPD. For this dataset, the peak of $L_vE$ is around 1200 UTC for both clear and cloudy skies although the peak for cloudy skies is less distinct due to the more fluctuating daily cycle of $Q^*$. Because $Q^*$ and $L_vE$ are out of phase, the highest values for $L_vE$ do not

occur in the bin with the highest net radiation, but rather in the bin of 400-500 W m⁻² (which roughly contains data from 11:00 UTC and after 12:00 UTC).

### 3.3.2 Cloud-induced radiation perturbations and response by turbulent fluxes

The short interval fluxes (one minute) of the double beam laser scintillometer (DBLS) technique enable us to study the vegetation response to rapid radiation perturbations due to changes in cloud cover. The goal here is to illustrate this potential

by discussing selected time-series under changing cloud conditions during IOP 2. The morning of IOP 2 was characterized by rapidly changing cloud conditions due to the overpass of a shallow cumulus cloud deck. A breakdown of the one-minute DBLS sensible heat flux in terms of contributions from turbulent exchange ($K_T$) and the measure for temperature fluctuations ($C_T^2$) is given in Fig. 11. This figure also depicts, on the same axes, scaled time-series of wind speed and PAR that can be regarded as proxies that fuel mechanically induced turbulence (wind speed) and buoyancy turbulence (radiation

in general) as well as photosynthesis (PAR).

First of all, the one-minute DBLS fluxes of H closely follow the cloud cover induced radiation changes, but with a time-lag of 45-120 seconds (Fig. 11a). This is similar to those reported by van Kesteren et al. (2013b). H fluxes measured with EC techniques even when estimated over the relatively short interval of ten minutes, which is not a standard output, are not capable of capturing such rapid dynamic behaviour of the flux regime (Fig. 11a). The dynamic behaviour in the DBLS H is

mainly governed by fluctuations in T expressed by $C_T^2$ (Fig. 11c) and to a lesser extent by changes in the exchange coefficient $K_T$ (Fig. 11b). Note that is impossible to fully distinguish the three variables H, $K_T$ and $C_T^2$ from each other as they are all inter-connected, e.g. $K_T$ is defined in terms of the Obukhov length L, which in turn depends on H and $u_*$. Nevertheless, our high-time-resolution observations demonstrate that changes in PAR induce very fast responses of the transported quantity T (Fig. 11c). Even in the absence of strong wind-induced variations in $K_T$, these T variations lead to

approximately similar dynamic behaviour of *H*. On top of this, the additional, but smaller wind induced fluctuations in $K_T$ are also reflected in H and lead to "noise" in the variability of H compared to the cloud-induced on-off behaviour of PAR.

Next we examine how soon the fluxes of $H_2O$ and $CO_2$ respond to the cloud induced radiation changes. Fig. 12 demonstrates that there is indeed a fast response, and the one-minute resolution fluxes of $H_2O$ and $CO_2$ allow us to precisely determine a

delay time of approximately two minutes for the increases $CO_2$ uptake and transpiration of $H_2O$ relative to the changes in PAR. The delay is once again undetectable with the standard thirty-minute eddy-covariance results (Fig. 12). This behaviour is in line with what was concluded about the state of the vegetation observed at leaf level (Sec. 3.1). As the vegetation is not water-stressed and is at a stage of development at which it is still actively growing, it will react rapidly to changes in

radiation, *i.e.* it is in a radiation-limited regime. Under the conditions of our study, stomata appear to have reacted only slowly or remained constantly open, because leaves were unstressed or reacting only slowly to cloud-induced changes. Moreover, the timescale of a light-induced stomatal response (maximum values twenty minutes, Van Kesteren, 2013b) is normally larger than the timescale of most fluctuations in radiation. Our suggested explanation is that the one- to two-minutes delay time observed between radiation and turbulent fluxes is due to processes associated to an inertia of the leaf in

addition to turbulent transport between the leaf and laser path due to e.g. the small but not negligible storage of heat, $H_2O$ and $CO_2$ in the canopy layer. However, we need further evidence to disentangle the separation in delays between $H_2O$ and $CO_2$ fluxes.

### 3.4 Sun-induced fluorescence (SIF) measurements: temporal variability

Studying spatial and seasonal variabilities in ET during plant growth was one of the key goals of CloudRoots. To this end,

we analysed SIF observations measured on time and on space. The top-of-canopy measurements of SIF were carried out in two ways: (i) diurnal courses from a single representative location were recorded from a stationary FLOX system, and (ii) mobile measurements covering several locations within a field were recorded from a FLOX system that was housed in a backpack. To ensure reproducible measurements the two fibre optics of the system were attached to a gimbal and were placed with a movable tripod 2 m above ground. Diurnal curves were acquired on 7 May, 4 and 14 June (only morning hours

due to cloudy conditions in afternoon); mobile measurements (with change of measurement locations during the day) on 6 June and 26 June. As SIF measurements should be performed under clear-sky conditions only, records affected by clouds were carefully removed. Aerial maps of SIF were acquired with the high-resolution imaging spectrometer HyPlant. Fig. 13a shows the aerial map of $F_{760}$ acquired on June 26[th], suggesting homogeneous canopy properties within the winter wheat study field, while great differences can be seen between different fields. The same image identifies the FloxBox measurement

locations in the same colour code that reconstruct the diurnal temporal variability of $F_{760}$ during the entire CloudRoots campaign in Fig. 13b.

Diurnal changes in photosynthetic activity are clearly visible in $F_{760}$. Measurements made at different locations generally follow the same diurnal pattern, especially within the period 7 May to 14 June, further confirming the hypothesis that ET spatial heterogeneity within the winter wheat field was small. The seasonal changes are also traced by $F_{760}$: From 7 May until

14 June, the winter wheat canopy was photosynthetically active in a transition stage from booting (7 May) until grain filling (14 June), as is reflected by high SIF values. At the end of June, however, the canopy approached senescence and the reduction in photosynthesis was documented by greatly reduced fluorescence levels (see Fig. 13b, see pink values after 12 UTC). This photosynthesis reduction is also corroborated by the normalised difference vegetation index (NDVI), which was

calculated as the normalized difference between far-red to red reflectance (see supplementary material for details). The green dense canopy has a NDVI value close to 1, and the decrease in NDVI is caused by the yellowish colour of the winter wheat canopy (see Fig. S2 at the supplementary information).

### 3.5 Connecting SIF and evapotranspiration flux at the landscape scale

It is difficult to directly quantify spatial variations in the ET flux with the currently available in-situ equipment due to the necessity of installing a large number of measurement stations. Recently some promising concepts have been published that exploit the relationship between SIF and plant water relations (Damm et al. 2018, Jonard et al. 2020). Following these concepts, we studied in two steps the connections between ET to regional measurements of SIF, which were recorded on this scale by the airborne sensor HyPlant (see Fig 13a). First, to obtain an estimation of the spatial variability ET at CloudRoots, we used the 15 km x 15 km map acquired by the HyPlant sensor on 26th June 2018 and a land use classification of the region (Lussem, 2018). ET cannot directly be measured, thus, it was predicted using different Kc coefficients that depend on the land use categories around CloudRoots. We define Kc as the ratio of ET over a particular crop relative to the ET of potential grass used as reference (Allen et al., 1998; Bogena et al., 2010). For this analysis, the regional land-use map that consisted of 32 different land-use classes was translated to a reduced classification scheme of 9 land-use classes, which covered most of the vegetation types in the study region (Table 4). Roads were excluded from the analyses, as we assumed that their effect is negligible on the 15 m x 15 m grid.

For the estimation of Kc ET coefficients, we used the plant developmental stage at the CloudRoots site at the end of June. For the main regional crops, namely sugar beet, winter wheat, winter barley, and potatoes, local measurements of evapotranspiration by EC towers were used. These data have been collected over several years and weekly averaged. This enabled us to compute Kc from measured and potential ET averaged over the last two weeks of June. In the particular cases of winter wheat and especially winter barley, the Kc coefficient changes rapidly at this time of the year, in extreme cases from 1.0 to 0.3 within two weeks, due to the onset of senescence. Therefore, the coefficients for these two crops shall be used with care. In absence of eddy-covariance data, we calculates the characteristic values of Kc for each crop type and the developmental stage were taken from Allen et al. (1998). All estimated Kc coefficients for different crops can be found in Table 4. To estimate the ET over a specific area occupied by particular crop on a given day and time, the land-use map was transferred to the map of Kc coefficients according to Table 4 and then multiplied by the potential ET, using the ET grass as a reference value (ETgrass), specific to that moment in time. Fig. 14 shows the spatial variability of predicted ET for the IOP 3 inferred from the Kc coefficients and the value of potential grass reference averaged between 09:00 and 14:00 UTC. The area is a 1 km x 1 km square, characterized by a mean of 5.76 mmol m$^{-2}$ s$^{-1}$ and a standard deviation of 1.86 mmol m$^{-2}$ s$^{-1}$. Fig. 14 shows that this method can provide plausible information on the variability of ET at the sub-kilometre scale and it points out to the need to introduce this sub-grid ET variability information in modelling studies. In the second step of the procedure, we compared this estimated ET to the SIF measurements (F$_{760}$). Fig. 15 shows the correlation between estimated ET and solar-induced fluorescence F$_{760}$ for 26th June (Julian day 177) for the different land covers. The correlation between

mean $F_{760}$ values and predicted ET values is $R^2 = 0.61$ with larger of ET and $F_{760}$ vales for crops and grass compared to the forest conditions. It is calculated from the comparison pixel by pixel of the SIF (Fig. 13a) and ET (Fig. 14.\). As the HyPlant overflight was carried out at noon in order to acquire the maximal SIF values and minimize the influence of changing sun angle, we also used the maximal value of ETgrass, measured at midday on 26th June. The large range of values of ET, $F_{760}$ and $F_{687}$ from the different land-use categories corroborate the large variability of ET around the CloudRoots field.

## 3.6 Boundary-layer integrated dynamics over heterogeneous landscapes

To integrate and improve the interpretation of our observations, we used CLASS to model the cloudless day 7 May 2018 (IOP 1). Our specific aims, related to the scales and processes under study, are: (i) at leaf level, to make use of the new constants in the mechanistic A-$g_s$ model obtained from the observations (Fig. 5 and Table 3), (ii) at landscape scale, to represent the sensible heat flux in a heterogeneous landscape and (iii) to estimate the potential impact of advection (heat) on the diurnal evolution of surface and boundary-layer variables. Table A2 summarises all initial and boundary conditions, constrained by the observations, which are employed in the modelling of the surface and atmospheric variables. Fig. 16 compares the model results with the surface and upper-air observations. Focusing first on Fig. 16a, we found that the modelled H largely overestimates the observations taken at the CloudRoots. However, comparing our modelled H with the estimate of the regional flux shown in Fig. 1b, we found a satisfactory agreement in terms of magnitude and diurnal variability between this regional observed flux and CLASS model calculation. Note that here, and compared to Table 4, we oversimplified the land-surface categories in two: "bare soil" and "vegetated". To complete this evaluation, we show in Figure S1 the impact of the optimized A-$g_s$ constants presented in Table 3 (CloudRoots) versus the default ones. Both, the evolution of surface fluxes and boundary-layer height are in better agreement with the observations. Similar impacts on how leaf processes (rice) can influence the meteorology were reported by Ikawa et al. (2018). There the boundary-layer temperature was changing up to 0.5 K depending on the constants used in the leaf photosynthesis model.

Our explanation of the improved comparison between the observations and the CLASS results using the aggregated sensible heat flux is the following: in a heterogeneous landscape such as the location of CloudRoots (Fig. 1a), each surface type contributes its own latent and sensible heat fluxes. It is the landscape aggregate of heat fluxes (named regional and shown with triangles in Fig. 16a and introduced in Fig. 1b), and more specifically the sensible heat flux, that governs the boundary-layer evolution in terms of height, potential temperature, specific humidity and atmospheric constituents. Only by using this higher H do we obtain satisfactory agreement with the observed boundary-layer height evolution, which reaches its maximum values at around 1500 m in the afternoon (Fig 8b).This further emphasises that the H measured with the EC instrument during CloudRoots is only representative of the specific measurement site (leaf and canopy scales). The landscape average is an aggregate of values of H made up of the mosaic of surfaces as shown in Fig.1. As a consequence, it is this composite H rather than, a local value of H, that is the main driver of the boundary-layer development (boundary-layer scales). With regard to ET, the model results are in good agreement with the local CloudRoots observations. This indicates the secondary and more local role played by ET in the dynamics of boundary layer development. For studies focusing on the

regional values of ET, it will be necessary to calculate landscape-scale aggregate following the same procedures as H, while for studies at the leaf and canopy scales the local observations of ET are representative. Focusing now on Fig 16b, we found a satisfactory agreement between the modelled boundary-layer height and the three independent observations made with three different instruments. In this Fig. 16b, it is interesting to note that the ABL height inferred by the radio sounding measurement collected more than 100 km distant from of the Cloud Roots site has values similar to those collected by the LIDAR located within a radius of 5 km from the CloudRoots site. We attribute these similar values to a boundary layer that is characterized by being spatial homogeneous and with a similar temporal evolution on the larger regional scale.

In CLASS, besides solving the diurnal variability of the boundary-layer dynamics and the state variables, offers the possibility of adding a large-scale contribution that represents the advection of heat and/or moisture (see Vilà-Guerau de Arellano et al., 2015). We have performed a sensitivity analysis to determine the role played by heat advection for the surface fluxes and the boundary-layer development. In the specific case that is modelled on 7 May, we relate this advection of heat or moisture to the diurnal evolution of H contrast between the measurement site and its adjacent fields, i.e. horizontal transport of heat, moisture or momentum is driven by secondary circulations induced by the different thermal characteristics of the fields around the CloudRoots site (Fig. 1a). More specifically, we prescribe an advective heat contribution to represent the horizontal transport of heat due to the thermal variability of the surface conditions. This term follows an exponential function (Table A2) with maximum positive values of advection equal to 0.9 K h$^{-1}$ at midday. This advective term is imposed only on the mixed-layer and not on the free troposphere. Fig. 16 shows how this advection of warm air to the CloudRoots site influences the boundary-layer height. Starting with H, warm advection leads to higher mixed-layer temperatures that reduce the gradient between the temperature at the surface and the atmosphere, and thus reduce H. We find an opposite effect on ET. The increase in temperature by advection of warm air leads to an increased atmospheric demand, and therefore enhances ET. With regard to the boundary-layer height, we might suppose that a drop in of H would lead to a decrease of the boundary-layer growth. However, the modelled boundary-layer height displays the opposite behaviour. This is because the lower H is partly offset by a decrease in the thermal inversion at the interface between the boundary layer and the free troposphere. Lower values of the difference in $\theta_v$ between the free troposphere and the mixed-layer enable boundary-layer air parcels to be more easily transported into the free troposphere, resulting in faster growth of the boundary-layer. This is because of the virtual potential temperature between the environmental and the parcel is effectively reduced. The CLASS model results show that this process is more important than the decrease in H at the surface, and it allows the boundary layer to grow deeper than in the numerical experiment in which the warm advection is omitted. These numerical sensitivity experiment analyses enable us to quantify how non-local processes, in particular the effects of the regional average H and of warm advection, influence the observations at the measurement site.

## 4. Discussion

CloudRoots offers an integrated methodology that combines field experiments across spatial scales (from leaf to landscape) closely linked to the modelling of the diurnal variability of the soil-plant-atmosphere continuum. To frame the discussion and link all our observations at the various scales and modelling efforts, we present in Fig. 17 all the different estimates of

ET obtained during the three IOPs, averaged between 09:00 and 14:00 UTC in order to avoid the morning and afternoon transitions. Plotted alongside the ET estimates, we showed the leaf-level measurement of $g_{sw}$ to indicate the control of vegetation on canopy-level ET. The four instrumental techniques are: sap flow, the eddy-covariance (EC), scintillometer (averaged over thirty minutes and one minute), ET inferred by the profile lift measurements and ET infrared from the SIF observations. The ET modelled by CLASS is also included for IOP 1.

In comparing ET from the three IOPs, we find significant differences in magnitude from different techniques. In general, the highest values of ET are observed during IOP 1. The three IOPs were characterized by differences in the stages of growth, from very active vegetation to senescent, and influenced by a range of weather conditions: IOP 1 cloudless, IOP 2 scattered and thick clouds, and IOP 3 shallow cumuli. It is surprising that the decay in the vegetation activity as quantified by the measurements of leaf conductivity (Fig. 3 lower panels) is less evident in differentiating IOP 3 (senescent stage) from the

more active vegetation at IOP 1 and 2. Furthermore we observed, moving from IOP 1 to IOP 3, a much stronger decline in $g_{sw}$, suggesting that stomatal closure compensated for increased atmospheric moisture demand.

Several conclusions can be drawn from this intercomparison of ET observations using different techniques. Firstly, we might expect that the EC/scintillometer measurements, both with larger footprint and the inclusion of the soil evaporation contribution, show a net total ET that is similar to or higher than that one obtained by the sap-flow measurements. Secondly,

we observed a far more pronounced response in declining $g_{sw}$ compared to all ET measurements. These results point to the need to measure more accurately the leaf energy balance to take the penetration of radiation in the canopy under clear and cloudy conditions into account. This would also require a revision of scaling procedure from the leaf to the canopy level. Secondly, it is known that the EC flux measurements normally underestimate the sensible and latent heat fluxes because the EC flux measurements filter out the low frequencies (Foken et al., 2008; Gao et al., 2017). This underestimation is difficult

to determine, but as a first-guess and related to Fig. 17 the underestimation might range between 10 and 15%.

Although the contribution of soil evaporation is small compared to plant transpiration due to the high vegetation cover, we need to stress that EC and scintillometer observations are similar to or smaller than the ET observed or inferred from the other techniques (Fig. 17). This highlights the difficulty of estimating ET due to the need to include and quantify the contributions of the four fundamental processes: soil evaporation, up-scaled leaf transpiration, evaporation related to the sap

flow and the two non-local processes, entrainment of dry air and horizontal advection of heat and moisture. Here, the modelling of ET, taking into account for and integrating all these processes, enables us to discriminate among these processes and calculate the budget of ET as a function of these local and non-local contributions. In that respect, the CLASS

model is a tool capable of efficiently combining observations and model results that integrate surface and boundary-layer dynamics. The averaged modelled ET is at the higher range of the ET observed estimations during IOP 1.

With respect to the differences between the one-minute and thirty-minute series measured by the scintillometer, their median is very similar in the three IOPs. However, differences become larger at smaller timescales due to the non-steadiness of ET under the presence of clouds. Here, the one-minute flux calculated from the scintillometer can capture the rapid and large fluctuations by clouds (Fig. 12), and in particular the maximum values. In order to obtain more definitive conclusions how ET varies under cloud conditions, we need to analyse in more detail other situations characterized by different diurnal cloud cycles, and systematically relate ET to key cloud characteristics such as the cloud optimal depth to determine how cloud thickness influences ET, and the time scale of the cloud passage.

Regarding the quantification of the different processes contributing to ET, Fig. 9 illustrates the need to continue to test analytical techniques to identify the individual contributions of soil and plants to determine the diurnal ET budget. A possibly useful tracer would be the stable isotopic composition of water vapour and carbon dioxide (Lee et al., 2009; Griffis 2013) and combined with isotope signals in modelling the surface and boundary-layer dynamics with the carbon and water exchanges. To further discriminate between soil and plant sources and sinks under unsteady conditions due to radiation and dynamic perturbations by cloud shading, these high-frequency stable isotope measurements should go beyond the typical average time of eddy-covariance (thirty minutes). As van Kesteren et al. (2013) showed, and is further corroborated in this work, the scintillometer technique combined with high-frequency observations of $H_2O$ and $CO_2$ enable us to quantify the responses time of ET and $CO_2$ assimilation to these intermittent radiation fluctuations or cloud flecks (Kaiser et al., 2018).

Finally, the integration of all processes in the CLASS model shows the challenges in interpreting the measurements taken at the sub-kilometre scales and adequately representing the surface turbulent fluxes. Although the measurements indicate that the day selected for the modelling displayed a very homogeneous boundary layer depth over an area with a radius of 100 $km^2$, the sensible heat flux measured at the CloudRoots facility was not representative of it. Therefore, recommend to extending the number of stations by means of a multi-tower approach that would also include also detailed observations of the soil and plant conditions. In addition to obtaining a more representative field sensible heat flux which is better related to the development of the boundary layer, a denser network of spatial observation stations is also necessary to estimate more accurately the role of hectometre-scale heterogeneity-induced circulations and their relationships with the local advection of heat and moisture (Mauder et al., 2010).

## 5. Conclusions

Our main findings, organised from the smaller to the larger scales observed and modelled, are summarized as follows:

- At *leaf scale*, we find that stomatal conductance and gross primary production decrease in line with the increasing senescence of the plant. The tiller-level measurements of the sap flow are virtually constant throughout the growing period. Underlying causes need to be further investigated under controlled conditions. The successful modelling of the leaf stomatal conductance and the photosynthesis assimilations required the relevant constants used in the mechanistic model (A-$g_s$) in the field to be measured. Modelled leaf-level photosynthesis compares better with the measurements during the mature growing period than during senescence. For future field experiments, we recommend of including leaf-level measurements in meteorological campaigns to improve calculations related to the water-carbon leaf and canopy exchanges.

- At *canopy scale*, the high frequency vertical profiles – measured in and above the canopy - of wind speed, potential temperature, specific humidity and carbon dioxide prove to be very valuable in obtaining profiles of gross primary production in the canopy and as a function of height. By inverting these observed profiles, we obtain an estimate of the contributions of soils and plants to the net evapotranspiration and $CO_2$ ecosystem exchange. The validation against individual measurements of these components gives better results for the net ecosystem exchange than those for the net evapotranspiration. We argue that for evapotranspiration the dependence on temperature and water vapor deficit plays a more important role than for $CO_2$ assimilation, the latter being mainly controlled by the partitioning between direct and diffuse radiation.

- Under *cloud conditions*, we show that the perturbation by clouds of direct and diffuse radiation create large fluctuations in evapotranspiration and the $CO_2$ assimilation with opposite signs for evapotranspiration and $CO_2$ exchange. A cloudy boundary layer reduces evapotranspiration, whereas it enhances plant assimilation of $CO_2$. The one-minute turbulent fluxes acquired by the scintillometer demonstrate the relevance of flux measurements observed at higher frequencies for improving quantification of the impact of clouds on the photosynthetically active radiation. With these fast-turbulent fluxes, we quantify delays of the turbulent fluxes with respect to the photosynthetically active radiation. These delays are on the order of minutes. Comparing these one-minute flux estimate with the standard thirty-minute average measured with the eddy-covariance technique, we find a lower median and a large increase in the variability of the net evapotranspiration. This information can be useful in determining the impact of rapid fluctuations driven by the impact of clouds on evapotranspiration and its impact on the closure of the surface energy balance.

- At *landscape and boundary-layer integrated scales*, the modelled sensible heat flux correlates better with the area-weighted average flux than the local flux estimates. The area-weighted flux integrates in a simple manner a composite of bare soil and vegetated surfaces at regional scale (kilometres). This aggregate regional flux is representative of an area that is larger than the CloudRoots site (100 m x 100 m). Therefore, a model setup that represents the boundary layer evolution well only needed to be informed by the area-weighted average of two main surface types, bare soil and vegetated areas. The variations of ET

due to surface heterogeneity were also measured and inferred from airborne sun-induced fluorescence observations. Our findings corroborate the large heterogeneity of ET at the sub-kilometre scales with values ranging from forest (about 2.5 mmol $m^{-2}$ $s^{-1}$) to late crops such as potato or sugar beet (8-10 mmol $m^{-2}$ $s^{-1}$) .

- The *comparison of all the ET measurements at the various scales* show that there are still large differences in observing ET among the different observing techniques, the modelling of ET and their relation to stomatal aperture during the entire growing season. These ET observations do not show a clear pattern related to the scale at which they were measured.

- The *modelling and scale integration* of this comprehensive observational data set enables us to study the carbon and water exchange at leaf, canopy and landscape levels. It also allow us to quantify how horizontal advection of heat within the mixed-layer influences the surface fluxes and the growth of the atmospheric-boundary layer. We show, for instance, that the horizontal advection of heat leads to deeper boundary-layer depths. This numerical experiment thus paves the way to more complete modelling studies, for instance using large-eddy simulation numerical experiments, on how surface and the overlaying atmosphere interact on sub-diurnal and sub-kilometre scales.

## 6. Author Contributions

JVG designed the CloudRoots study and approach. OH and AG designed and coordinated the CloudRoots field experiment. The individual measurements were gathered by: OH (scintillometer), HB (stomatal aperture/photosynthesis), AK and AG (elevator profiles), ML (sap flow), MM and DA (SIF), TR, GA, NB and YR (stable carbon and water vapour isotopes). HB integrated the observations to be connected with the modelling efforts. GG, GM, HB and JVG performed the numerical experiments with CLASS. PN and JVG wrote the paper with key contributions from all the authors: OH, HB, KD, DE, GG, AK, ML, MM, GM, AM, UR, TR, GA and AF.

## 7. Data availability

All CloudRoots observations are archived at https://www.tr32db.uni-koeln.de and the search term "cloudroots"). The data can be obtained upon request at Only CRC/TR32 participants are allowed to apply for an account. Please contact the TR32DB admin for further information. CLASS model (Python and Fortran versions) is freely available at http://classmodel.github.io/.

## 8. Acknowledgements

This study was financed by the Deutsche Forschungsgemeinschaft (DFG) Collaborative Research Centre 32 (TR32) "Patterns in Soil-Vegetation-Atmosphere System". The contribution of AG and deployment of the profile elevator, microlysimeter and part of the isotope measurements were financed by the German Federal Ministry of Education and

Research (BMBF) within the framework of the project "IDAS-GHG" (Grant: FKZ 01LN1313A). Airborne acquisition and data analysis with the HyPlant sensor were financed by the European Space Agency (ESA) in the frame of the FLEXSense campaign (ESA Contract No. ESA RFP/3-15477/18/NL/NA). The contribution of PN to the analysis were financed by the European Space Agency (ESA) within the project PhotoProxy (Grant. ESA RFP/3-15506/18/NL/NF). The ancillary hardware and its maintenance were supported by TERENO (https://www.tereno.net). The authors wish to thank Bernhard Pospichal and Tobias Marke for contributing measurements on boundary layer properties. These data were provided by the Jülich Observatory for Cloud Evolution (JOYCE-CF), a core facility funded by DFG via grant DFG LO 901/7-1. Hubert Hüging and Huu Thuy Nguyen from the INRES Crop Science Group of Bonn University for operating the sap-flow equipment.

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

# Figures

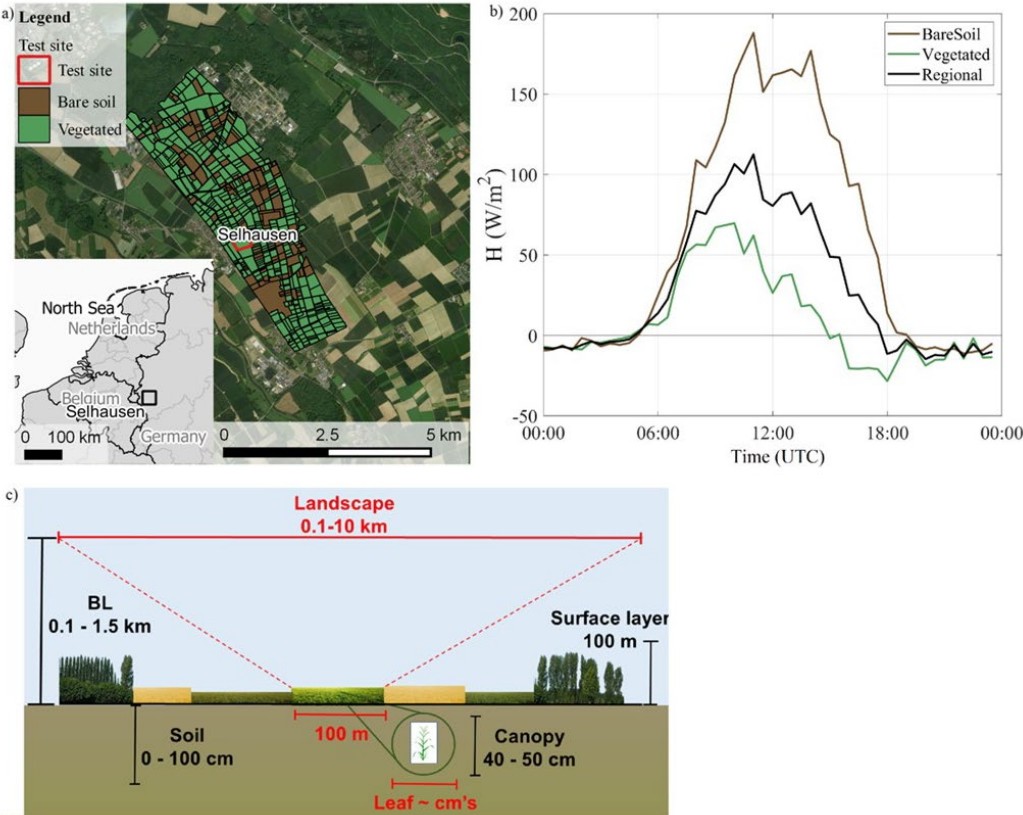

**Figure 1:** a) Aerial view (Bing Maps, © 2019 Microsoft Corporation © 2019 DigitalGlobe © CNES (2019) Distribution Airbus DS) of the observation area. The ICOS Selhausen test site is located in the middle of the 10 x 10 km map section. The surrounding agricultural area was classified into the categories bare soil (including "late crops" after Table 3) and vegetated ("early crops", forest and grassland after Table 3) during the IOP 1. b) Corresponding sensible heat flux (H) during IOP 1, whereby H of bare soil and vegetated area were measured and the regional average was estimated as weighted average (60% and 40% for vegetated and bare soil, respectively). c) Schematic sketch of horizontal (red) and vertical (black) length scales influencing the measurements. The larger indicated horizontal and vertical scales indicate the spatial scales of boundary layer dynamics. Horizontally, the 100 m scale is the size of the field hosting the ICOS test site.

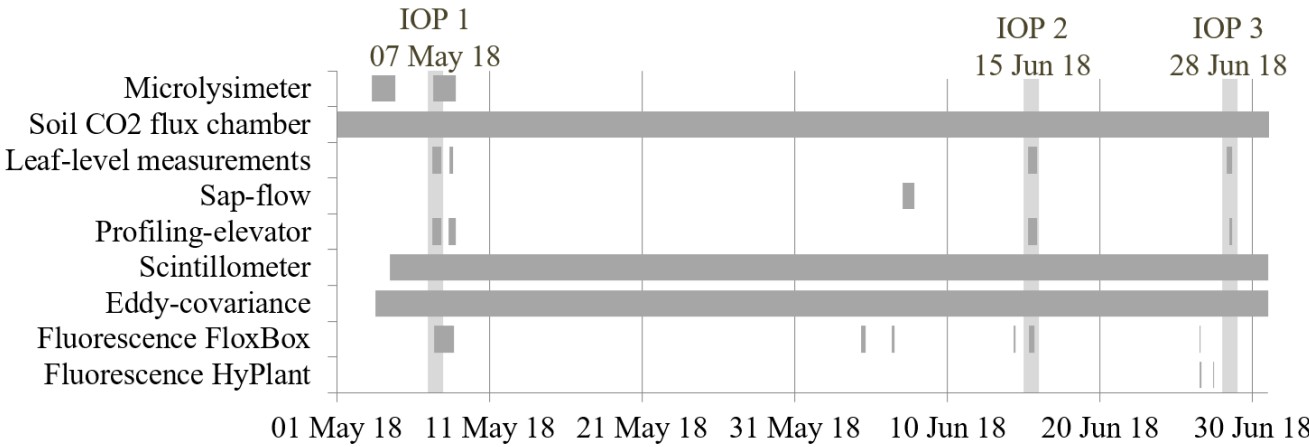

**Figure 2:** Campaign-specific measurement setup and temporal developments from May to June 2018, including three intensive operation periods (IOP).

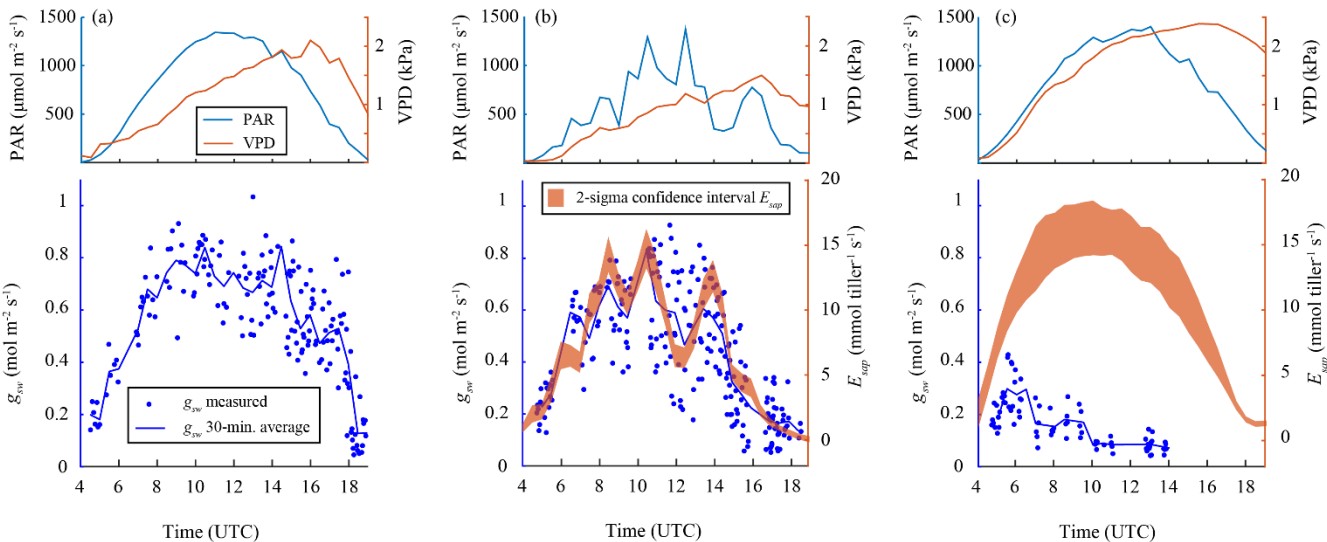

**Figure 3:** Upper panels: diurnal changes in photosynthetically active radiation (PAR) and vapour pressure deficit (VPD) measured for (a) IOP 1, (b) IOP 2 and (c) IOP 3. Lower panels: Leaf-level measurements of stomatal conductance of water vapour ($g_{sw}$), in b) and c) compared to tiller-level measurements of sap flow ($E_{sap}$). Leaf-level measurements of $g_{sw}$ (blue markers) were averaged over thirty-minute intervals (blue line). Sap flow measurements represent the one-standard-deviation confidence interval (shaded region) of measurements on 24 tillers averaged over 30-minute time scales.

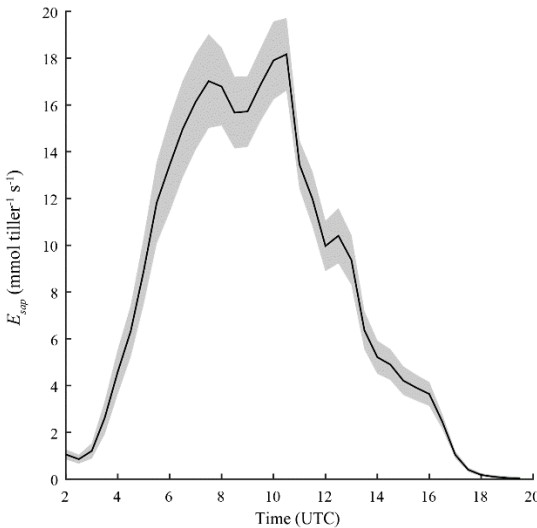

**Figure 4:** Sap-flow measured using the heat-balance method for 7 June 2018 (non-IOP day).

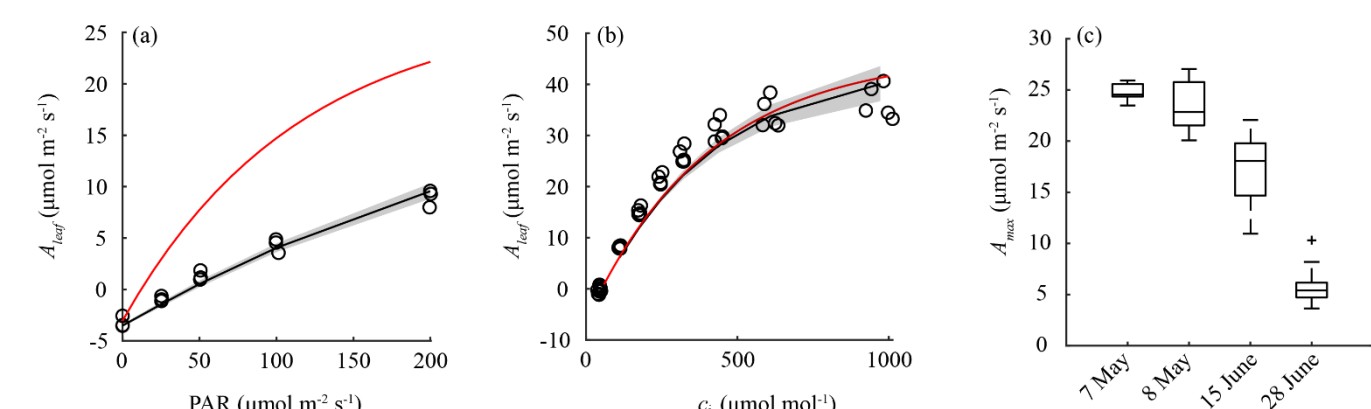

**Figure 5:** Measurements of leaf-level photosynthesis ($A_{leaf}$) as function of photosynthetically active radiation (PAR) (a) and leaf-interior $CO_2$ concentrations ($c_i$) (b). These results were used to parameterize the A-$g_s$ model for IOP1, as indicated by the black line and shaded one-standard-deviation confidence interval. The red line indicates the model response using the default parameter values. (c) Observed and modelled seasonal decline in maximum light-saturated photosynthesis ($A_{max}$). Boxes indicate the variability in observed values in $A_{max}$, red markers indicate the modelled net photosynthesis rate using fitted values for $A_{m,max298}$. Fitted and default A-$g_s$ model parameter values are indicated in Table 3.

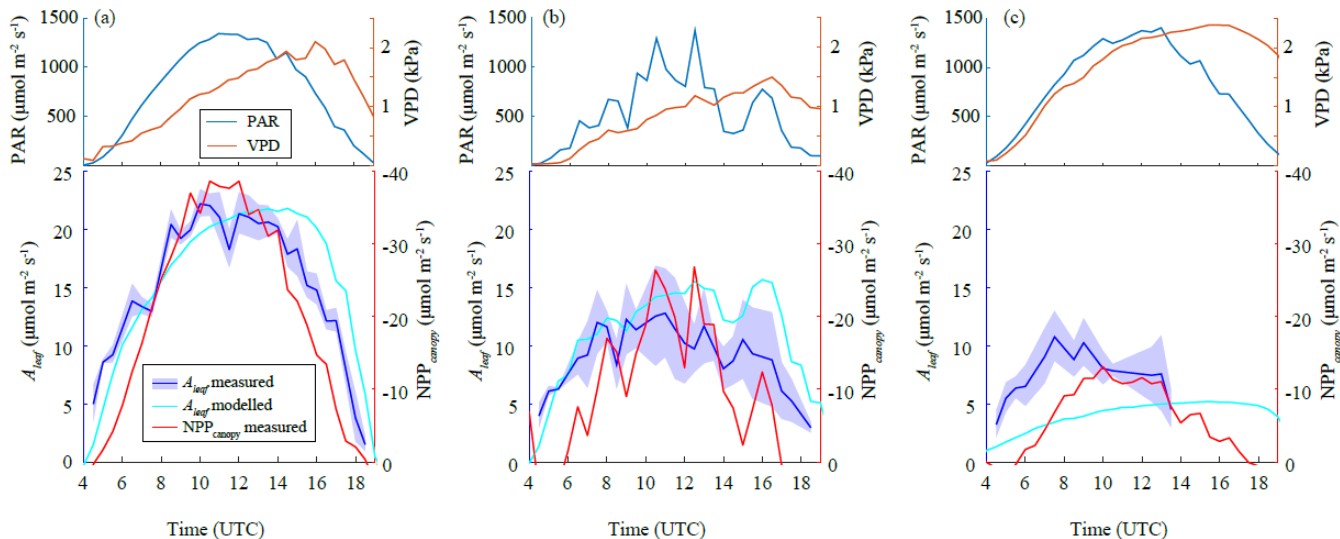

**Figure 6:** Measured leaf-level photosynthesis ($A_{leaf}$) compared to modelled $A_{leaf}$ using the A-$g_s$ model and canopy-level net primary productivity ($NPP_{canopy}$) for (a) IOP 1, (b) IOP 2 and (c) IOP 3. Measurements of $A_{leaf}$ were plotted as 30-minute averages (blue line) and their one-standard-deviation confidence interval (shaded region). Upper panels show diurnal changes in photosynthetically active radiation (PAR) and vapour pressure deficit (VPD) measured for each IOP.

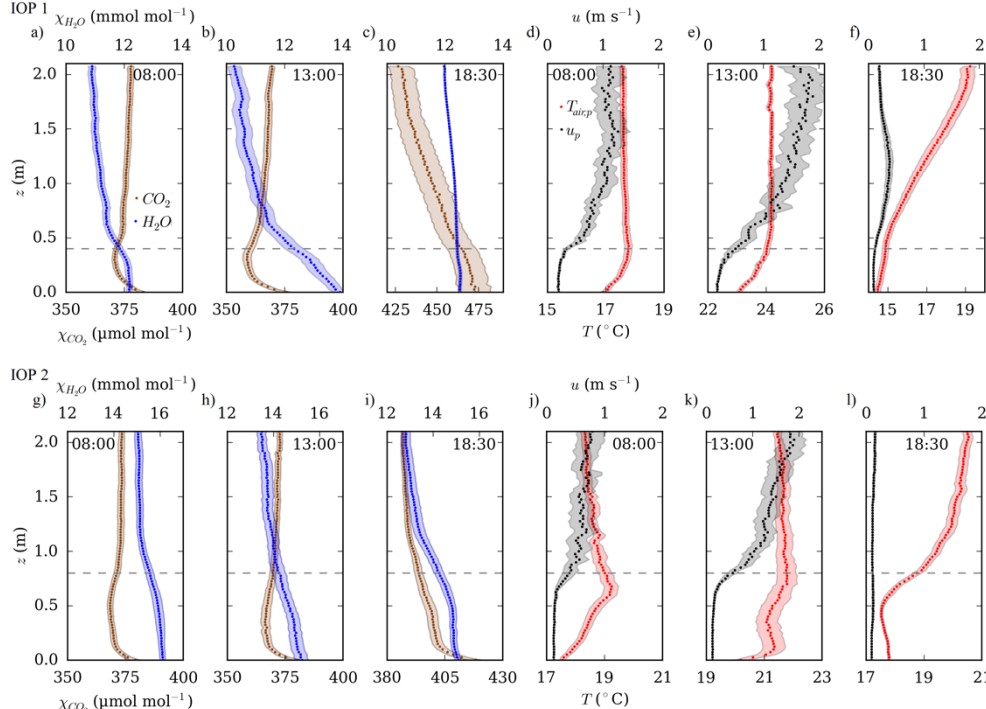

**Figure 7:** Selected (08:00, 13:00 and 18:30 UTC) 30-minutes mean profiles of the $H_2O$ and $CO_2$ mole fractions ($\chi H_2O$, $\chi CO_2$), wind speed ($u_p$) and temperature ($T_{air,p}$) measured at high vertical resolution during IOP 1 (upper panel) and IOP 2 (lower panel). Shaded areas indicate the 95 % confidence interval resulting from the standard deviation between individual profiles sampled within a thirty-minute average interval. The dashed lines indicate the canopy heights.

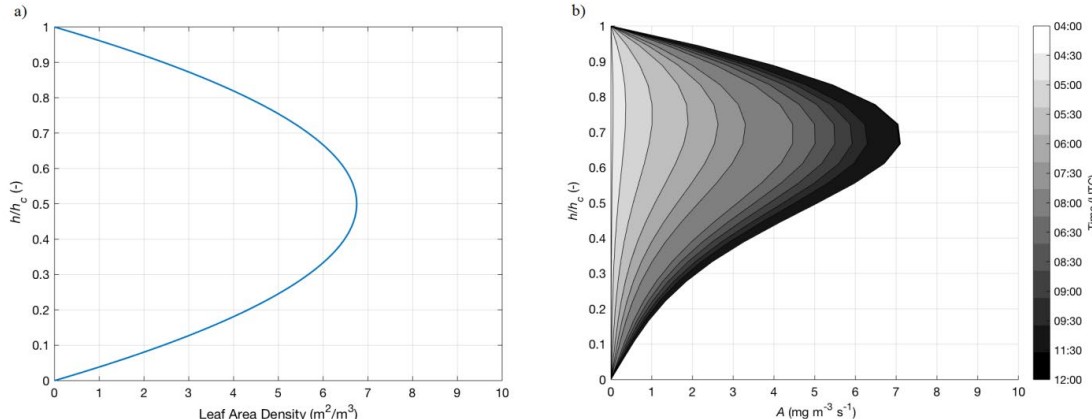

**Figure 8:** (a) Leaf area density ($m_{leaf}^2$ m$^{-3}$) on 7 May 2018 as a function of height (h) normalized to the maximum canopy h (h$_c$). The profile is typical for winter wheat as defined by Olesen et al. (2004). (b) Time evolution of $CO_2$ gross primary production A (mg m$^{-3}$ s$^{-1}$) on 7 May 2018 as function of h normalized to h$_c$. The profile is obtained using the profile measurements and using Eq. (2).

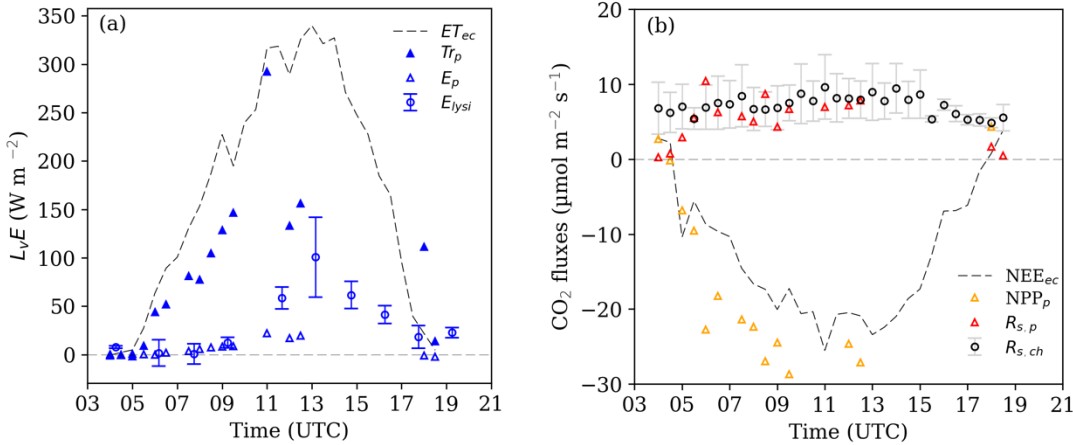

**Figure 9:** Source partitioning results for (a) $H_2O$ and (b) $CO_2$ fluxes for IOP 1. Grey dashed lines show the measured latent heat flux (ET$_{ec}$) and net ecosystem exchange (NEE$_{ec}$) in half-hourly time steps. Values with subscript index $p$ indicate estimate based on inversed profile concentration measurements (Sec. 3.4). Error bars for evaporation calculated from microlysimeters (E$_{lysi}$) and soil respiration measurements (R$_{s,ch}$) indicated to one standard deviation. (ET$_{ec}$: evapotranspiration measured as latent heat flux L$_v$E by the eddy-covariance system; $E$: evaporation; Tr: transpiration, NPP: aboveground net primary production; R$_s$: soil respiration).

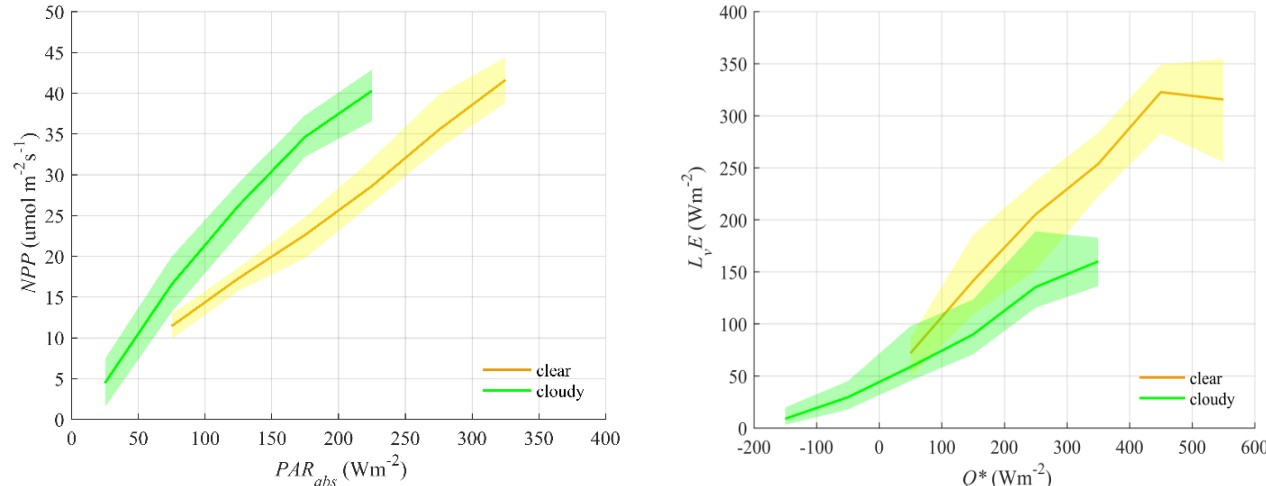

**Figure 10:** Left panel: net primary production (NPP) versus the photosynthetic active radiation (PAR). Right panel: Latent heat ($L_vE$) versus net radiation ($Q^*$). In both figures, the observation period encompasses clear and cloudy skies during a two week-period starting on 7 May 2018 at 03:30 UTC (sunrise) and ending on the 20 May 2018 at 19:40 UTC (sunset). The solid line represents the median of the data. The lower and upper boundaries of the shaded area are the 25[th] and 75[th] percentiles of the data respectively.

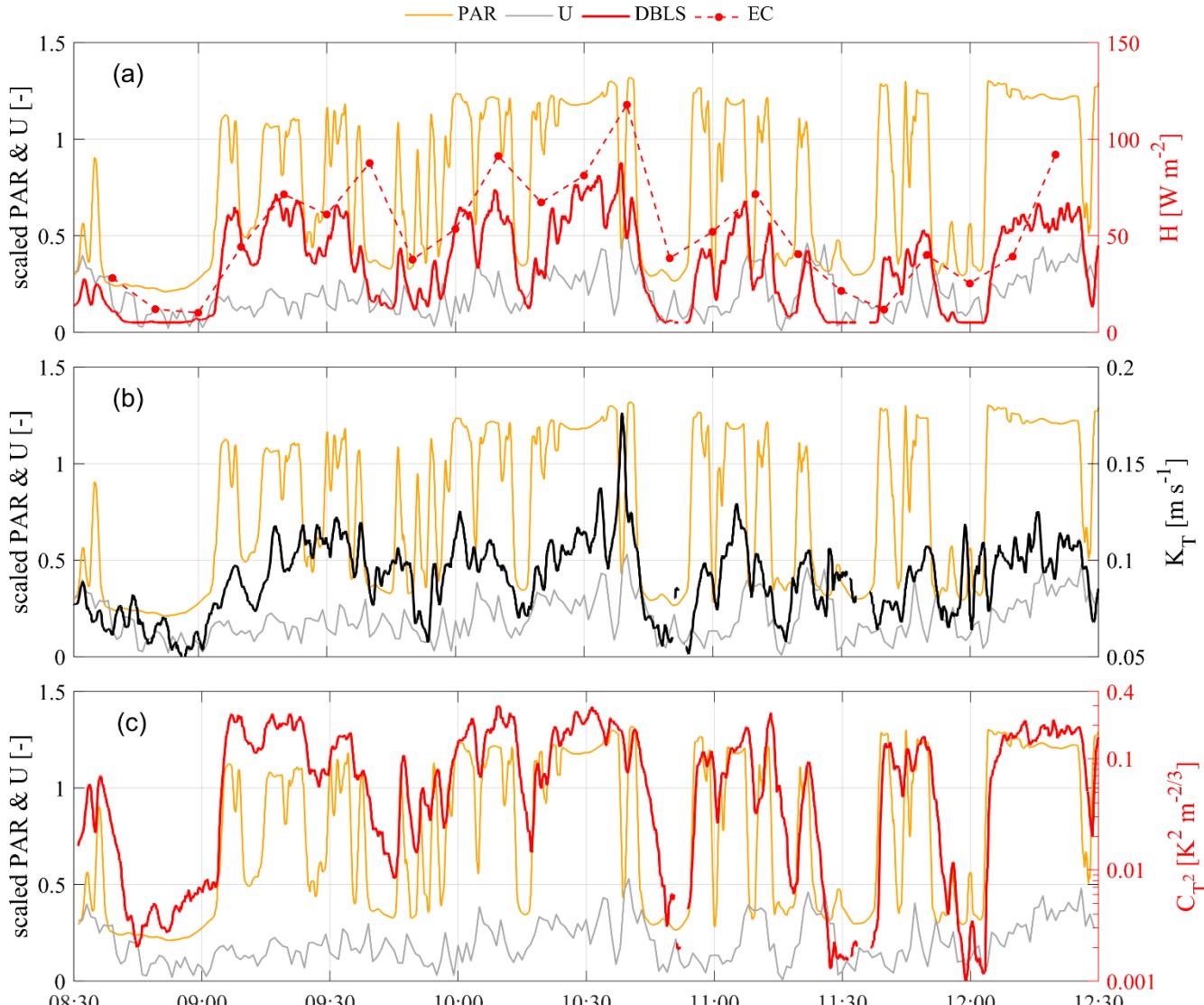

**Figure 11:** IOP 2 (15 June 2018) time-series of: (a) sensible heat flux (H) at one-minute intervals with a displaced beam laser scintillometer (DBLS) and at ten-minute intervals with an eddy-covariance system (EC), combined with scaled time-series of photosynthetically active radiation (PAR, scaled by 1500 μmol m$^{-2}$ s$^{-1}$) and wind speed (U, scaled by 6 ms$^{-1}$); (b) turbulent exchange coefficient $K_T$ and (c) structure parameter of temperature, $C_T^2$ that together make up H in the DBLS method following Eq. (1).

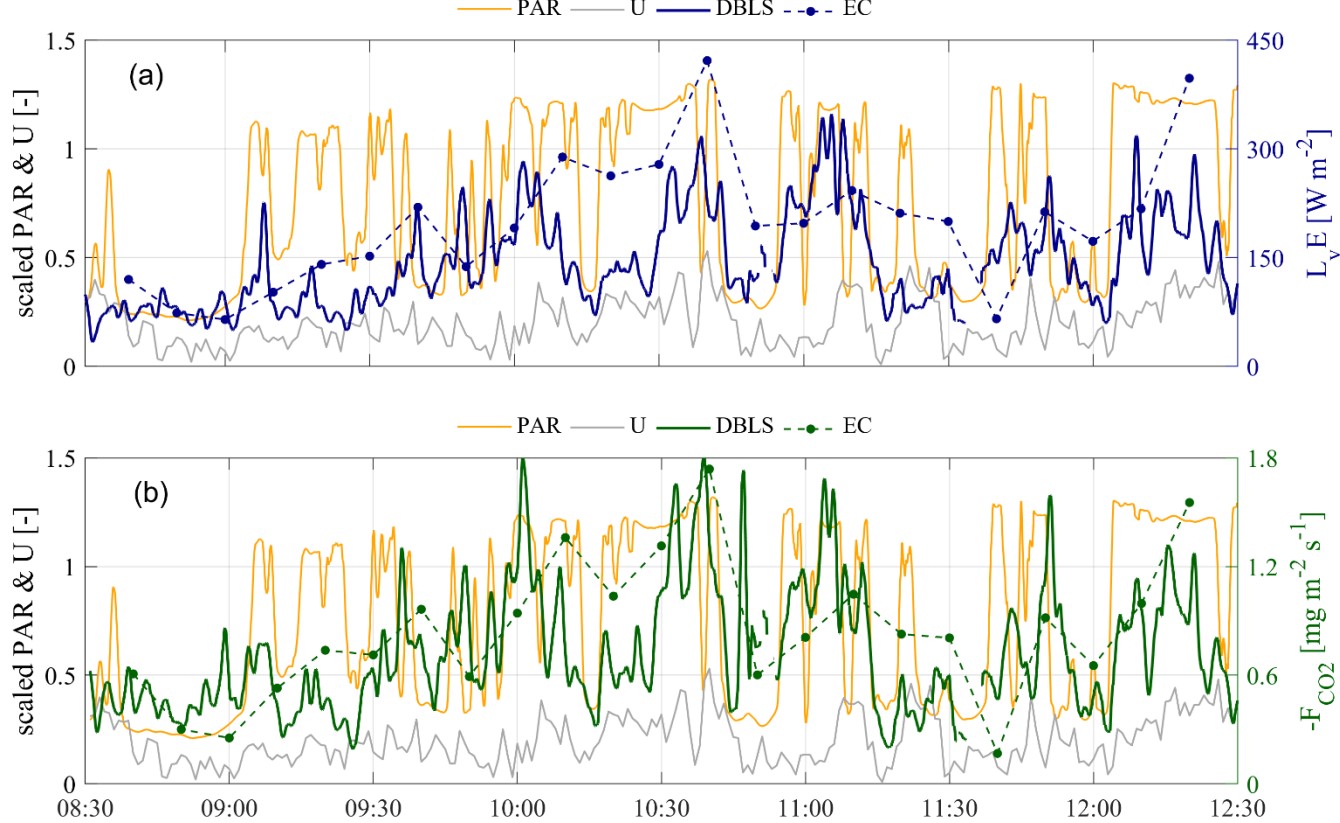

**Figure 12:** IOP 2 (15 June 2018) time-series of: (a) latent heat fluxes (LvE) at one-minute intervals with a displaced beam laser scintillometer (DBLS) and at ten-minute intervals with an eddy-covariance system (EC) combined with scaled time-series of photosynthetically active radiation (PAR, scaled by 1500 μmol m-2 s-1) and windspeed (U, scaled by 6 ms-1); (b) same as (a) but for the CO2 flux ($F_{CO2}$).

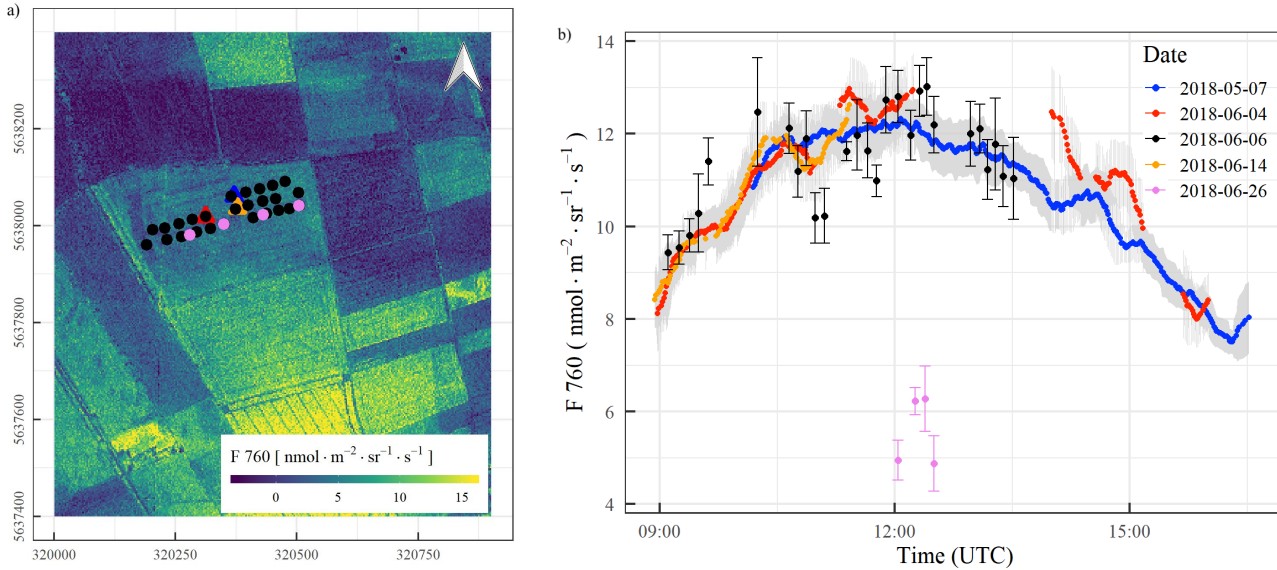

**Figure 13:** a) Aerial map of $F_{760}$ on 26 June 2018 with measurement locations used to combine with mobile (circles) and stationary (triangles) measurements. b) Diurnal changes in $F_{760}$ on different days of the campaign as five-minute measurement averages depicted in the same colours as observation locations in a).

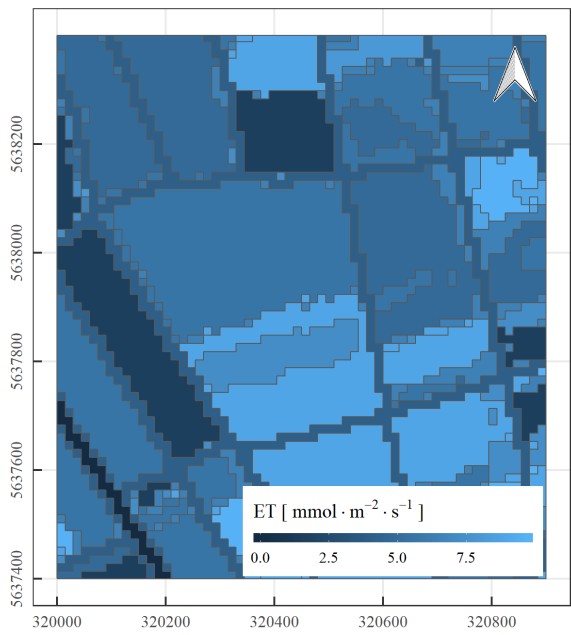

**Figure 14**: Spatial variability of evapotranspiration inferred from combining $K_c$ coefficients with the value of potential grass reference ETgrass. The x- and y-axis represent the geographical coordinates of the CloudRoots site in metres (50°51'57.3"N 6°26'42.5"E).

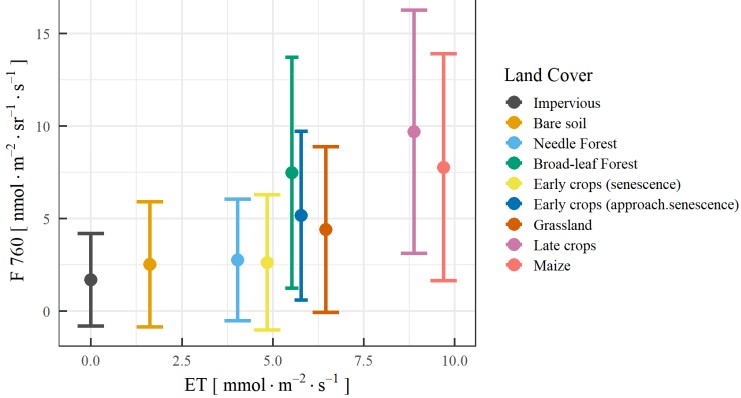

**Figure 15**: Relation between evapotranspiration and fluorescence F760 including the standard deviation for the nine land-use categories defined in Table 3. The data were collected on 26 June 2018.

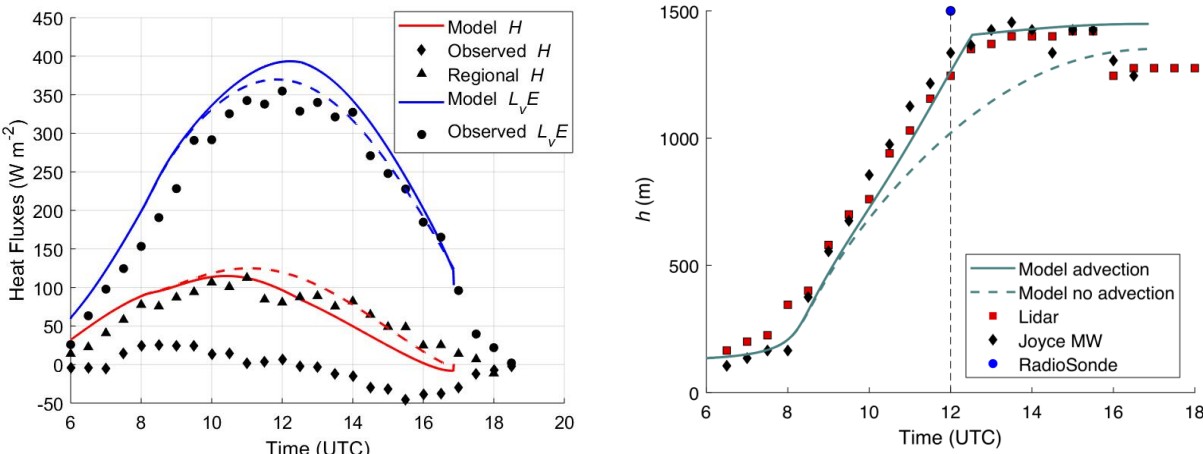

**Figure 16:** Comparison of the model and observed results of 7 May 2018: (left) surface fluxes and (right) boundary-layer depth. The regional H, an aggregate that combines the vegetated and bare soil surfaces around the CloudRoots site as shown in Fig. 1b, is also included. For the boundary-layer depth estimations, we used three different observational techniques. The LIDAR and microwave (MW) techniques were located at the JOYCE site facility. Solid and dashed lines represent the model results of surface fluxes and boundary-layer height with and without imposing the advection of heat, respectively (Table A2) for complete the information on initial and boundary conditions.

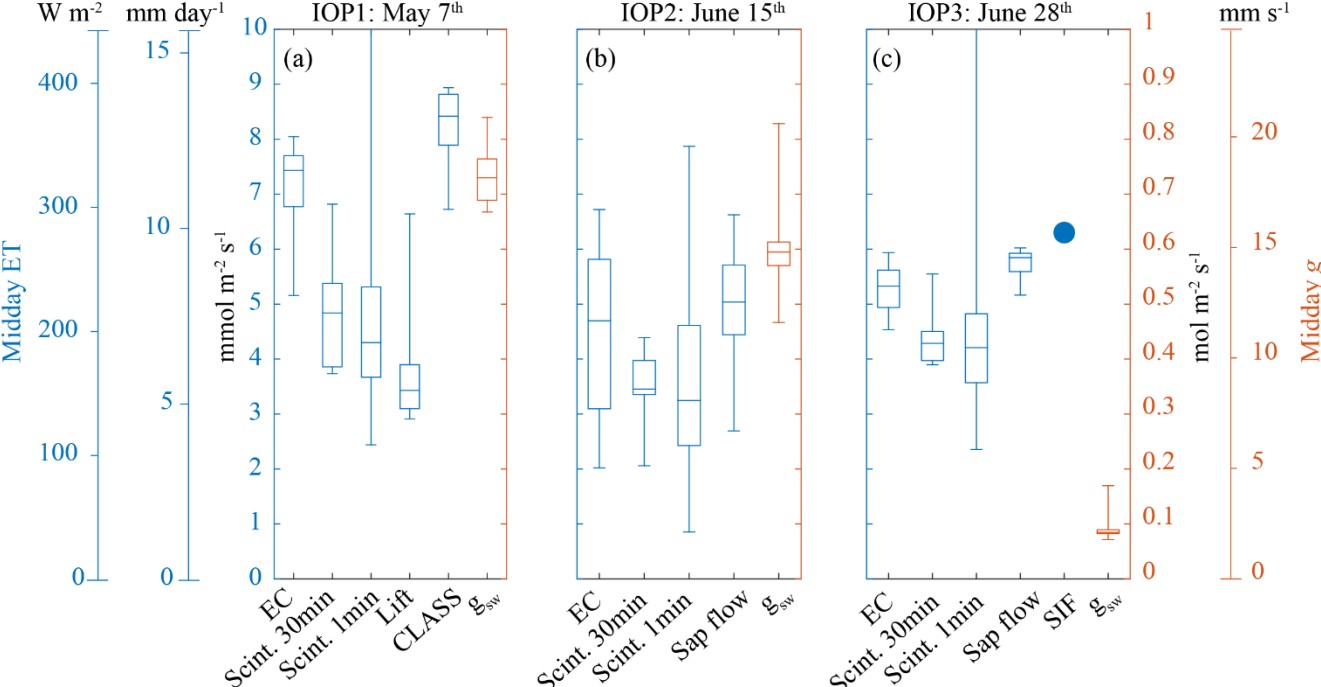

**Figure 17**: Summary of midday evapotranspiration collected using different instrumental techniques during (a) IOP 1, (b) IOP 2 and (c) IOP 3. ET fluxes (left y-axis) and $g_{sw}$ (right y-axis) reflect the period from 09:00 to 14:00 UTC. Box plots denote the variability in thirty-minute measurement intervals, except for the one-minute scintillometer measurements.

5 Central mark of each box indicates the median, and the bottom and top edges of the box indicate the 25th and 75th percentiles, respectively. The acronyms are eddy-covariance (EC), scintillometer (Scint) with thirty-minute (30min) and one-minute (1min) averages, ET inferred from the lift profiles (Lift), sap flow, ET calculated with the CLASS model and ET inferred from the sun-induced fluorescence (SIF).

**Table 1:** Meteorological and biometric conditions during the intensive operation periods on 7 May (IOP 1), 15 June (IOP 2) and 28 June 2018 (IOP 3). Global radiation, water vapour-pressure deficit (VPD), photosynthetically active radiation (PAR) and soil water content (SWC) are daily averages. The meteorological variables were measured at the height 2.4 ± 0.1 m (see Section 2.3.7 for details).

|  | IOP 1 | IOP 2 | IOP 3 |
|---|---|---|---|
| **Meteorological conditions** |  |  |  |
| cloud amount | 0-1 | 3-6 | 0-4 |
| temperature range (°C) | 7.0 - 25.4 | 13.2 - 23.9 | 10.1 – 27.6 |
| wind range (m s$^{-1}$) | 0.1 – 2.1 | 0.06 – 1.5 | 0.2 – 3.3 |
| global radiation$^*$ (W m$^{-2}$) | 514 | 311 | 462 |
| **Biometric conditions** |  |  |  |
| canopy height (m) | 0.45 | 0.80 | 0.78 |
| LAI (m$^2$ m$^{-2}$) | 4.5 | 5.5 | 5.5 |
| VPD / VPD$_{max}$ (hPa) | 11.7 / 20.9 | 7.6 / 14.9 | 16.0 / 23.6 |
| PAR$^*$ (μmol m$^{-2}$ s$^{-1}$) | 768 | 475 | 741 |
| SWC 5, 20, 50 cm (vol.%) | 0.20 / 0.27 / 0.30 | 0.17 / 0.19 / 0.22 | 0.12 / 0.15 / 0.21 |

5   $^*$ Daily averages calculated from sunrise to sunset

**Table 2:** List of symbols, description, units and the representatively scale.

| Symbol | Description | Unit | Scale represented |
|---|---|---|---|
| A | photosynthesis rate | μmol m$^{-2}$ s$^{-1}$, mg m$^{-2}$ s$^{-1}$ | landscape |
| A$_{leaf}$ | leaf-level photosynthesis rate | μmol m$^{-2}$ s$^{-1}$, mg m$^{-2}$ s$^{-1}$ | leaf |
| A$_m$ | maximum light-saturated photosynthesis | μmol m$^{-2}$ s$^{-1}$, mg m$^{-2}$ s$^{-1}$ | landscape |
| A$_{m, max298}$ | maximum leaf-level photosynthesis rate | μmol m$^{-2}$ s$^{-1}$, mg m$^{-2}$ s$^{-1}$ | leaf |
| E | evaporation | mm, W m$^{-2}$ | several |
| E$_{lysi}$ | evaporation from microlysimeters | W m$^{-2}$ | landscape |
| E$_p$ | evaporation profile based | W m$^{-2}$ | leaf |
| E$_{sap}$ | sap flow | μmol tiller$^{-2}$ s$^{-1}$ | leaf |
| ET | evapotranspiration | mm, W m$^{-2}$ | several |
| ET$_{ec}$ | evapotranspiration eddy-covariance | W m$^{-2}$ | canopy |
| g$_{sw}$ | stomatal conductance of water vapour | mol m$^{-2}$ s$^{-1}$ | leaf |
| h | height | m | boundary-layer |
| h$_c$ | canopy height | m | canopy |
| L | Obukove lenght | m | canopy |
| L$_v$E | latent heat flux | W m$^{-2}$ | several |
| LAD | leaf area density | m$^2$ m$^{-3}$ | canopy |
| LAI | leaf area index | m$^2$ m$^{-2}$ | canopy |

| | | | |
|---|---|---|---|
| NEE | net ecosystem exchange | $\mu mol\ m^{-2}\ s^{-1}$, $mg\ m^{-2}\ s^{-1}$ | canopy |
| $NEE_{ec}$ | net ecosystem exchange eddy covariance | $\mu mol\ m^{-2}\ s^{-1}$, $mg\ m^{-2}\ s^{-1}$ | canopy |
| $NPP/NPP_{canopy}$ | net primary production | $\mu mol\ m^{-2}\ s^{-1}$, $mg\ m^{-2}\ s^{-1}$ | canopy |
| $NPP_p$ | net primary prodyuction inferres profile | $\mu mol\ m^{-2}\ s^{-1}$, $mg\ m^{-2}\ s^{-1}$ | canopy |
| PAR | photosynthetically active radiation | $\mu mol\ m^{-2}\ s^{-1}$, $W\ m^{-2}$ | leaf/canopy |
| Q* | net radiation | $W\ m^{-2}$ | leaf/canopy |
| $R_d$ | $CO_2$ dark respiration | $mg\ m^2\ s^{-1}$ | landscape |
| $R_s$ | soil respiration | $\mu mol\ m^{-2}\ s^{-1}$ | landscape |
| $R_{s,ch}$ | soil respiration measured by chamber | $\mu mol\ m^{-2}\ s^{-1}$ | landscape |
| $R_{s,p}$ | soil respiration inferred from profile | $\mu mol\ m^{-2}\ s^{-1}$ | landscape |
| S↓ | global radiation | $W\ m^{-2}$ | landscape↓ |
| H | sensible heat flux | $W\ m^{-2}$ | canopy/landscape |
| T | temperature | °C, K | several |
| $T_{air}$ | air temperature | °C, K | landscape |
| $T_{air,p}$ | air temperature from vertical profile meas. | °C, K | leaf/canopy |
| $T_{rp}$ | transpiration, profile-based | $W\ m^{-2}$ | leaf/canopy |
| u | wind speed | $m\ s^{-1}$ | landscape |
| $u_p$ | wind speed from vertical profile meas. | $m\ s^{-1}$ | landscape |
| $u_*$ | friction velocity | $m\ s^{-1}$ | landscape |
| VPD | water vapour-pressure deficit | kPa | leaf/canopy |
| α | light-use efficiency | $mg\ J^{-1}$ | landscape |
| $α_0$ | initial value of light-use efficiency | $mg\ J^{-1}$ | landscape |
| $\chi H_2O$ | mole fractions of $H_2O$ concentration | $\mu mol\ mol^{-1}$ | leaf/Canopy |
| $\chi CO_2$ | mole fractions of $CO_2$ concentration | $\mu mol\ mol^{-1}$ | leaf/Canopy |

**Table 3:** Parameters representing the maximum leaf-level photosynthesis rate ($A_{m,max298}$) and the initial value of light-use efficiency ($\alpha_0$) under low light, as adjusted in the original A-$g_s$ model to represent plant-specific photosynthesis characteristics for winter wheat (ww). $A_{m,max298}$ was initially fitted using the A-Ci curves and $\alpha_0$ is fitted using the A-PAR curves taken during IOP 1 (Fig. 5). For IOP 2 and IOP 3, $A_{m,max298}$ values were fitted only on leaf-level measurements of $A_{max}$. The values of IOP 1 were used as numerical settings for the CLASS model runs (Fig. 16). The equivalence to typical values of the commonly used in the Farquhar-Berry-von Caemmerer (FBvC) model of leaf photosynthesis (Farquhar *et al.*, 1980) is given in Table S1 at the supplementary information.

| Fitted model variable | Default value (for C3 plants) | Fitted ww IOP 1 | Fitted ww IOP 2 | Fitted ww IOP 3 |
|---|---|---|---|---|
| Mesophyll conductance at 298 K [mm s$^{-1}$] | 7.0 | 10.0 | 10.0 | 10.0 |
| maximum leaf-level photosynthesis rate ($A_{m,max298}$) [mg m$^{-2}$ s$^{-1}$] | 2.2 | 1.926 | 1.0 | 0.2 |
| light-use efficiency ($\alpha_0$)[mg J$^{-1}$] | 0.017 | 0.0053 | 0.0053 | 0.0053 |

**Table 4:** Estimated $K_c$ coefficients for different land-use classes, which are dominant in the study area. The land-use classes were calculated using a more detailed land use classification that consisted of 32 classes. For this study several classes having similar transpiration rates were combined.

| | Land-use class | Kc | Main surface types included |
|---|---|---|---|
| 1 | Impervious | 0.0 | Roads, urban areas, industrial areas |
| 2 | Bare soil | 0.2 | Bare fields, incl. harvested fields with rapeseed harvest residuals |
| 3 | Needle forest | 0.5 | Managed spruce and pine forest |
| 4 | Broad-leaf forest | 0.7 | Broad-leaf forest, scrubs |
| 5 | Early crops (senescence) | 0.6 | Winter barley |
| 6 | Early crops (approaching senescence) | 0.7 | Winter wheat |
| 7 | Grassland | 0.8 | Natural grasslands, urban grasslands |
| 8 | Late crops | 1.1 | Sugar beet, potato |
| 9 | Maize | 1.2 | Maize |

**Appendix 1**

**Construction of light and radiation response curves under clear and cloudy conditions**

For the construction of the light and radiation responses curves in Fig. 10, the data were divided into bins of PAR and $Q^*$.

For Fig 10a, we divide the data points in bins of incoming total PAR. Each bin covers a range of 50 Wm$^{-2}$, starting at 0-50 Wm$^{-2}$ and presumably ending at around 350-400 Wm$^{-2}$ (maximal intensity for incoming direct PAR). For Fig 10b, and the variable $Q^*$ each bin covers a range of 100 W m$^{-2}$, starting at -200 to 100 W m$^{-2}$ for cloudy skies and at 0 to 100 W m$^{-2}$ for clear skies. In both figures, for each data point the diffuse fraction of PAR is determined by combining measurements of incoming total PAR and incoming diffuse PAR. Subsequently, a data point is labelled 'clear' for diffuse fractions < 0.3 and

labelled 'cloudy' for diffuse fractions > 0.8. We choose these boundaries to balance a distinct difference between clear and cloudy skies with a large enough sample size for each bin. For clear skies, the first two bins are missing. This is due to the fact that under clear skies low levels of $Q^*$ are the result of the sun being close to the horizon, and as a result solar radiation has to travel a long distance through the atmosphere before reaching the surface. In those cases, most of the solar radiation reaches the surface as diffuse radiation due to *Rayleigh* scattering and scattering by aerosols, and therefore does not meet the

criteria to be labelled "clear". For cloudy skies, bins are missing for high levels of $Q^*$. Clouds attenuate solar radiation through absorption and backscattering, and thereby reduce $Q^*$ to a level lower than it would be for a clear sky.

**Table A1:** Management activities on the test site over the winter wheat cultivation cycle before, during and after the observation period of the CloudRoots campaign.

| Date | Management | Product |
|---|---|---|
| 25 Oct 2018 | sowing crop seeds | winter wheat (Premio) |
| 8 Mar 2018 | fertilisation | 81.6 kg N ha$^{-1}$ |
| 9 Apr 2018 | herbicide treatment | 200 g ha$^{-1}$ Broadway |
| 9 Apr 2018 | herbicide treatment | 1 l ha$^{-1}$ CCC720 |
| 22 Apr 2018 | fertilisation | 39.2 kg N ha$^{-1}$ |
| 2 May 2018 | fungicide treatment | 1 l ha$^{-1}$ Capalo |
| 2 May 2018 | fungicide treatment | 0.3 l ha$^{-1}$ Corbel |
| 2 May 2018 | herbicide treatment | 0.3 l ha$^{-1}$ CCC720 |
| 16 May 2018 | fertilisation | 50 kg N ha$^{-1}$ |
| 19 May 2018 | fungicide treatment | 1.5 l ha$^{-1}$ Adexar |
| 19 May 2018 | fungicide treatment | 0.5 l ha$^{-1}$ Diamant |
| 19 May 2018 | insecticide treatment | 0.3 l ha$^{-1}$ Bulldock |

| 16 July 2018 | harvesting | winter wheat, 92 dt ha[-1] |
| 19 July 2018 | Straw pressed and removed | |
| 25 Aug 2018 | ploughing | |
| 18 Sep 2018 | harrowing | |

**Table A2:** Initial and boundary conditions prescribed in CLASS to reproduce IOP 1 (7th May 2018).

| Mixed-layer model parameters | | |
|---|---|---|
| **Parameter (units)** | **Value** | **Source** |
| time steps (s) | 60 | - |
| runtime (s) | 50400 | - |
| residual-layer starting height (m) | 135 | Joyce microwave |
| surface layer top height (m) | 1400 | radiosonde |
| surface pressure (Pa) | 100600 | EC pressure gauge |
| large-scale wind divergence (s$^{-1}$) | 0 | default |
| $f_c$ (m s$^{-1}$) | $1 \cdot 10^{-4}$ | latitude |
| Coriolis parameter (-) | 0.2 | default |
| *Potential temperature* | | |
| initial mixed-layer temperature (K) | 286.2 | profile data and radiosonde |
| jump in potential temperature from boundary layer to free troposphere (K) | 4 | radiosonde |
| jump in potential temperature from boundary layer to residual layer (K) | 4.4 | radiosonde |
| free troposphere lapse-rate for potential temperature (h < 1400 m) (K) | $4.9 \cdot 10^{-3}$ | radiosonde |
| free troposphere lapse-rate for potential temperature (h < 1400 m) (K) | $6.2 \cdot 10^{-3}$ | radiosonde |
| advection of heat into the mixed-layer (K s$^{-1}$) | | $2.5 \cdot 10^{-4} e^{-\frac{(t[UTC]-12)^2}{5}}$ |
| *Specific humidity* | | |
| initial function mixed-layer specific humidity (kg kg$^{-1}$) | | $0.0067 - 0.0004(t[UTC] - 6.5)$ |
| Residual-layer lapse rate for specific humidity (kg kg$^{-1}$ m$^{-1}$) | $-1.4 \cdot 10^{-3}$ | radiosonde |
| free troposphere lapse-rate specific humidity (h < 1400 m) (kg kg$^{-1}$ m$^{-1}$) | $-2.7 \cdot 10^{-6}$ | radiosonde |
| free troposphere lapse-rate specific humidity (h < 1400 m) (kg kg$^{-1}$ m$^{-1}$) | $-9.0 \cdot 10^{-6}$ | radiosonde |
| advection of specific humidity into the mixed-layer (kg kg$^{-1}$ m$^{-1}$) | 0 | default |
| *Carbon dioxide* | | |
| initial mixed-layer $CO_2$ ($\mu mol CO_2$ mol$^{-1}$air) | 400 | profile measurements |
| jump in $CO_2$ at the inversion layer ($\mu mol CO_2$ mol$^{-1}$air) | -44 | profile measurements |
| free troposphere lapse-rate for $CO_2$ ($\mu mol CO_2$ mol$^{-1}$air m$^{-1}$) | 0 | default |
| advection of $CO_2$ into the mixed-layer ($\mu mol CO_2$ mol$^{-1}$air s$^{-1}$) | 0 | default |
| *Wind* | | |
| initial wind speed in the longitudinal direction (m s$^{-1}$) | 1.75 | profile measurements |
| jump in longitudinal wind velocity at the inversion layer (m s$^{-1}$) | 3 | profile measurements |

| | | |
|---|---|---|
| free troposphere lapse-rate for longitudinal wind velocity (m s$^{-1}$ m$^{-1}$) | $-1.8 \cdot 10^{-3}$ | profile measurements |
| advection of longitudinal wind into the mixed-layer (m s$^{-1}$ s$^{-1}$) | 0 | default |
| wind speed in the latitudinal direction (m s$^{-1}$) | 0 | default |
| jump in latitudinal wind velocity at the inversion layer (m s$^{-1}$) | 0 | default |
| free troposphere lapse rate for latitudinal wind velocity (m s$^{-1}$ m$^{-1}$) | 0 | default |
| advection of latitudinal wind into the mixed-layer (m s$^{-1}$ s$^{-1}$) | 0 | default |
| | | |
| roughness length for momentum (m) | 0.02 | canopy height |
| roughness length for scalars (m) | 0.002 | canopy height |

*Geographical coordinates and radiation*

| | | |
|---|---|---|
| latitude (deg) | 50.9 | geographical location |
| longitude (deg) | 6.4 | geographical location |
| Julian day-of-year (days) (7 May 2018) | 127 | data selected case |
| start time (hrs UTC) | 6.0 | - |
| cloud cover fraction (-) | 0 | camera |
| cloud-top radiative divergence (W m$^{-2}$) | 0 | camera |

*Soil*

| | | |
|---|---|---|
| soil moisture top soil layer (m$^3$ m$^{-3}$) | 0.177 | soil measurements |
| soil moisture deep soil layer (m$^3$ m$^{-3}$) | 0.286 | soil measurements |
| Vegetation cover fraction (-) | 0.98 | visual inspection, camera |
| T top soil layer (K) | 285.5 | soil measurements |
| T deep soil layer (K) | 284 | soil measurements |
| Clapp & Hornberger parametre a (-) | 0.219 | soil composition |
| Clapp & Hornberger parametre b (-) | 5.3 | soil composition |
| Clapp & Hornberger parametre p (-) | 4 | soil composition |
| saturated soil conductivity for heat (-) | $3.56 \cdot 10^{-6}$ | soil composition |
| saturated volumetric water content (-) | 0.472 | soil composition |
| field capacity volumetric water content (-) | 0.3 | soil composition |
| wilting point volumetric water content (-) | 0.154 | soil composition |
| parameter to calculate top layer soil moisture tendency (-) | disabled | soil composition |
| parameter to calculate top layer soil moisture tendency (-) | disabled | soil composition |
| LAI (-) | 4.5 | on-site determination |
| correction factor transpiration for VPD for high vegetation (-) | 0 | vegetation height |
| minimum soil resistance [s m$^{-1}$] | 50 | default |
| albedo (-) | 0.2 | radiation measurements |
| surface temperature (K) | 286.3 | profile measurements |
| thickness of water layer on wet vegetation (m) | 0.0002 | default |
| equivalent water-layer depth for wet vegetation (m) | 0.0001 | on-site observations |
| thermal conductivity skin layer | 5.9 | default |

| **A-g$_s$ model parameters** | | |
|---|---|---|
| CO$_2$ compensation concentration (mg m$^{-3}$) | 68.5 | C3 reference value |
| function parameter to calculate CO$_2$ compensation (-) | 1.5 | C3 reference value |

| | | |
|---|---|---|
| mesophyll conductance (m s$^{-1}$) | 10.0 | leaf gas exchange |
| maximum assimilation rate for $CO_2$ at 298 K (mg m$^{-2}$ s$^{-1}$) | 1.926 | leaf gas exchange |
| reference temperature to calculate mesophyll conductance (K) | 278 | C3 reference value |
| reference temperature to calculate mesophyll conductance (K) | 301 | C3 reference value |
| function parameter to calculate maximal primary productivity (-) | 2.0 | C3 reference value |
| reference temperature to calculate maximal primary productivity (K) | 281 | C3 reference value |
| reference temperature to calculate maximal primary productivity (K) | 311 | C3 reference value |
| maximum value of the ratio between the leaf and external (-) | 0.89 | C3 reference value |
| regression coefficient to calculate the ratio between the leaf and external $CO_2$ concentration (-) | 0.07 | C3 reference value |
| initial low-light-conditions use efficiency for $CO_2$ (mg J$^{-1}$) | 0.0053 | leaf gas exchange |
| extinction coefficient PAR (m m$^{-1}$) | 0.7 | C3 reference value |
| minimum cuticular conductance (mm s$^{-1}$) | $2.5 \cdot 10^{-4}$ | C3 reference value |