# Peer review of "CloudRoots: Integration of advanced instrumental techniques and process modelling of sub-hourly and sub-kilometre land-atmosphere interactions"

_Biogeosciences, 2020_

## Referee Comment (RC1) · Anonymous Referee #1 · 12 May 2020

Review for Vila-Guerau de Arellano

The authors proposed a concept of CloudRoots to investigate CO2 and H2O transfers from a leaf scale to the regional scale, connecting results from various techniques of different spatial scales. Te authors' intention to bridge researches from multiple disciplines at different scales together is novel because doing so is essential for further understanding land-atmosphere interaction. Many interesting results and insights are introduced across the manuscript, and I suppose the work will give a great contribution to land-atmosphere interaction research. Let me provide some comments that hopefully improve the manuscript.

**Page 1**

L20 (and related to photosynthesis parameters)

I suppose the necessity of accurate leaf parameters to describe photosynthesis and stomatal conductance is such an obvious fact, and I am afraid if it can be any interesting finding. Instead, can the authors provide any insights with respect to how accurate leaf parameters are important for ABL? Leaf parameters greatly vary even among C3 plants (e.g., Miner et al., 2017), and it is quite common to see a global-scale terrestrial ecosystem model using different leaf parameters by different vegetation types (e.g., Sun et al., 2014). One of most widely used sun/shade photosynthesis model by de Pury and Farquhar (1997) already manifests the importance of N effects on a photosynthetic capacity, and that may explain a large part of the differences in Am and alpha0 between the campaigns listed in Table 3.

If the authors discuss how tuned leaf parameters affects ABL, I tempt to suggest looking at a parameter that determines stomatal conductance in the A-gs model in addition to two parameters already introduced. I suppose f0 is the one (P247 in Vilà-Guerau de Arellano et al., 2015), and it can be determined by the outputs of LI6400. For example, Ikawa et al., (2018) reports a leaf parameter for stomatal conductance ($m_l$ in their paper in Table 4) of rice considerably affects ABL temperature.

L25 resolution of?

L26 non-linear behavior to what?

L28 please spell out ET when introduced for the first time

L29 inferred

**Page 2**

L2 evapotranspiration of total ET?

L9 Isn't "stomatal responses" a part of "surface and boundary-layer dynamics"?

L17 Similarly, "wind" is a part of boundary-layer dynamics? It may be better to replace "BL dynamics"

with other specific terms.

L22 first is redundant

L24 How large is a grid of weather and climate models?

**Page 3**

L10 diurnal variability of CO2-H2O flux partition

L17 first is redundant

L24 evapotranspiration or ET? Are they different?

**Page 6**

L5 mm instead of mm m-2, assuming 1 gram of H2O is 1 cubic centimeter.

[0.1g]/[(0.2^2*pi/4)m2] = [0.1cm3] /[(0.2^2*pi/4)m2] = 0.1/ (0.2^2*pi/4)/1000 mm

L21 were

L23 which leaf? Fully expanded? Or totally random in a canopy?

L28 Please use the SI unit instead of ppm for CO2 concentration.

**Page 7**

L1 Parameterization based on a rapid A-Q curve depends on stomatal conductance. I advise to include an average gsw value during the measurement of PAR = 0 – 200 umol m-2s-1.

https://www.licor.com/env/support/LI-6800/topics/rapid-light-curve.html

L14 superscript -2

**Page 8**

L15 Considering the expertise of the authors, I will leave it to them whether to call it MO length or O length.

http://glossary.ametsoc.org/wiki/Obukhov_length

L15-16 Please check the sentence.

L22 Please check the sentence.

**Page 9**

L11 2.105 may be too precise.

**Page 10**

L22 were

L5 what surface? Plant-canopy?

L19 Fig. 5

L24 Table 3 and it is already mentioned in L20.

**Page 12**

L27 Without knowing local time (or solar time), it is difficult to catch up discussion.

**Page 13**

L6 Fig. 3

L10 How was latent heat flux from EC? Was it also high in IOP3 despite small gsw?

L18 internal $CO_2$ concentration

L25 Fig. 5

**Page 14**

L10 subscript 2

L20-21 The presence (and not absence) of the local maxima increases thermal stability?

**Page 15**

L7 Please check the unit

L16-17 Please note that detail profile does not necessarily provides accurate values of gross primary production at least given our limited knowledge of plant canopy micrometeorology, though the profile approach is still useful for understanding mechanisms (e.g., Drewry et al., 2014).

L19 soil respiration or Rs?

L26-27 Ep?

L27 What is the sub-optimal performance?

**Page 16**

L25 It is a quite interesting topic that LE and $CO_2$ flux behave differently under diffuse and direct radiations. Can $CO_2$ flux also be included in Fig 10 to ensure the opposite pattern? It may be worth looking at a bulk stomatal conductance estimated from latent heat flux to delineate the effects of conductance and VPD (e.g., Dolman et al., 1991).

**Page 17**

L6 reaches

L7 I prefer to know this information of time earlier.

**Page 20**

I am afraid 3.4.2 needs a substantial improvement in the clarification. I was totally lost in the latter part of 3.4.2 and not able to understand how ET was estimated from SIF in Fig 16. I was not also able to understand what information authors want to convey by Fig 15. What is the correlation? It does not look like a correlation coefficient (0-1), and I do not see any relationships between x and y-axis either.

**Page 21**

L4 delete "it"

L23 H or SH? Also some are italic and some are not.

L27 Can it be simply because that the greater the net heat received by both horizontal and vertical fluxes (surface flux and advection), the faster the ABL height grows?

**Page 22**

L26 I apologize in advance for the lack of knowledge in the operational theory of scintillometer, but is the scintillometer measurement free from the concern of low frequency component? The spatial scale of 86.8 m over 60 secs by the Eulerian measurements may be still smaller than the low-frequency scales that would not be captured by EC measurements (greater than a spatial scale that would be inferred by the scale of tower height, wind speeds and time periods of 30 mins)?

**Page 24**

P24L16 those for the

**Page 25**

L1 It sounds contradicting to the importance of using accurate leaf parameters. Maybe the variations of crop coefficients (K) are so small among different vegetation covers (0.7-1.1) compared to the difference between vegetative and non-vegetative (0 – 0.2) areas (Table 4)?

**Figs and Tables**

Fig 4 Check the number of the fig.

Fig. 7 I assume the dashed lines are canopy heights.

Fig. 16 Was this ET estimated by SIF? I still did not understand how it was estimated.

Table 1 Height of wind measurements?

Table 2 Please check units.

Table 3 Which was determined by A-PAR and which was by A-Ci?

**Reference**

de Pury, D.G.G., Farquhar, G.D., 1997. Simple scaling of photosynthesis from leaves to canopies without the errors of big-leaf models. Plant Cell Environ. 20, 537–557. https://doi.org/10.1111/j.1365-3040.1997.00094.x

Dolman, A.J., Gash, J.H.C., Roberts, J., Shuttleworth, W.J., 1991. Stomatal and surface conductance of tropical rainforest. Agric. For. Meteorol. 54, 303–318. https://doi.org/10.1016/0168-1923(91)90011-E

Drewry, D.T., Kumar, P., Long, S.P., 2014. Simultaneous improvement in productivity, water use, and albedo through crop structural modification. Glob. Change Biol. 20, 1955–1967. https://doi.org/10.1111/gcb.12567

Ikawa, H., Chen, C.P., Sikma, M., Yoshimoto, M., Sakai, H., Tokida, T., Usui, Y., Nakamura, H., Ono, K., Maruyama, A., Watanabe, T., Kuwagata, T., Hasegawa, T., 2018. Increasing canopy photosynthesis in rice can be achieved without a large increase in water use-A model based on free-air $CO_2$ enrichment. Glob. Change Biol. 24, 1321–1341. https://doi.org/10.1111/gcb.13981

Sun, Y., Gu, L., Dickinson, R.E., Norby, R.J., Pallardy, S.G., Hoffman, F.M., 2014. Impact of mesophyll diffusion on estimated global land $CO_2$ fertilization. Proc. Natl. Acad. Sci. 111, 15774–15779. https://doi.org/10.1073/pnas.1418075111

Vilà-Guerau de Arellano, J., van Heerwaarden, C.C., van Stratum, B.J., van den Dries, K. van den, 2015. Atmospheric Boundary Layer: Integrating Air Chemistry and Land Interactions. Cambridge University Press, New York.

---

## Referee Comment (RC2) · Dennis Baldocchi (Referee) · 24 Jun 2020

CloudRoots: Integration of advanced instrumental techniques and process modelling of sub-hourly and sub-kilometre land-atmosphere interactions Jordi Vilà-Guerau de Arellano1

Biogeosciences

Understanding sub grid variability of mass and energy fluxes, especially over hetero-geneous landscapes or those subjected to full sun and shade under days with fair

weather clouds remains a challenge to our field. This is an interesting effort to bring together observations and models to better understand these processes. I've always been curious about how sunlight can drive fluxes which in turn drive pbl growth and humidification of the boundary layer, which then affect humidity and the production of fair weather clouds. Hence a mix of positive and negative feedbacks acting in concert.

This paper describes the detailed CloudRoots experiment. It joins a class of papers, like many of the key papers in the past that described FIFE, HAPEX, BOREAS, the Boardman ARM Field Experiment and various Kansas studies that provide the background for large investigator integrated field experiments. This newer generation of studies has some advantages ove the past studies with the emergence of SIF as a proxy for photosynthesis. Plus there are better sensors for boundary layer height and more flux stations.

Overall it has the connections between leaf and soil, to canopy, to landscape, to region to boundary layer set of measurements and models to provide a rich database for discovery, model validation and model parameterization.

Methods The team is using state of art eddy covariance measurements methods that are well vetted, though ICOS. Spatial integration is with scintillometers and inference of fluxes with remote sensing like SIF. The CLASS model is one of the best and is based on LES origins and can be used to drive a simpler one dimensional model. The scale of this work is over 10 km, which is reasonable. The study was in Germany and conducted as 3 intensive field experiments over the summer growing season. Land is flat and the landscape is well documented and assessed. Interesting to see mini lysimeters, soil CO2 chambers and leaf physiological capacity measured, eg leaf gas exchange and sap flow, too. Well planned and executed.

My one complaint is use of MOST to interpret scintillometer measurements. They are often advocated to measure spatial averages, but the edges of the sampling will see advection, so there can be problems inverting fluxes with 2D measurements into a 1D

framework. Eddy covariance works around this by establishing an internal boundary layer with large fetch.

One thing I do like about scintillometer, and something this team has done, is look at instant fluxes with sun and shade and calculate changes in surface conductance. That approach to me has proven powerful and interesting.

Van Kesteren, B., Hartogensis, O. K., Van Dinther, D., Moene, A. F., De Bruin, H. A. R., & Holtslag, A. A. M. (2013). Measuring H2O and CO2 fluxes at field scales with scintillometry: Part II–Validation and application of 1-min flux estimates. Agricultural and forest meteorology, 178, 88-105.

All this information is then integrated through measurements of boundary layer development. Not sure of other such comprehensive field studies, that compete, even FIF, BOREAS or HAPEX.

Results

Information on plant physiological performance is pertinent as it provides parameter information for subroutines in CLASS. I'd like to know more about Vcmax, Jmax and the Ball Berry or Medlyn/Leuning type stomatal conductance parameters used by the model.

The paper plays a new role to look at cloud induced fertilization of evaporation. Most studies focus on enhancement of co2 flux. So this is new and novel.

Interesting to see

At constant Q*, the median of LvE is always higher under clear skies than for cloudy skies. Diffuse fraction plays a minor role and the decrease on LvE under cloudy conditions is mainly due to the reduction in theincoming shortwave radiation.

There remains some debate and discussion on the term evaporation over evapotranspiration. I favor the former after hearing John Monteith advocate for it.

The paper drives towards a connection with large scale integrated SIF to landscape average evaporation. Since water and carbon fluxes are tightly coupled, I am open minded to this spatial scaling assessment.

At boundary-layer integrated scale, they find that modelled sensible heat flux correlates better with the area weighted average flux than the local flux estimates. I find this interesting as we are dealing with a similar problem. How to best average fluxes from a network of flux towers as a lower boundary condition for a pbl model? Or do we get integration of fluxes at 30 m scale with ECOSTRESS?

This leads me to a suggestion. The PI should also look at ECOSTRESS data for their domain and compare the integrated evaporative fluxes with what they are producing. These data are publicly available and could be a nice alternative constraint.

Fisher, J. B., et al. (2020), ECOSTRESS: NASA's Next Generation Mission to Measure Evapotranspiration From the International Space Station, Water Resources Research, 56(4), e2019WR026058, doi:10.1029/2019wr026058.

---

## Author Comment (AC1) · 9 Jul 2020

Response to reviewer RC1

Answer: We thank reviewer 1 for the thorough reading and his/her comments. We have taken the majority of his/her comments into account. Below (and in blue) we have provided a point-to-point responses.

The authors proposed a concept of CloudRoots to investigate $CO_2$ and $H_2O$ transfers from a leaf scale to the regional scale, connecting results from various techniques of

**BGD**

different spatial scales. The authors' intention to bridge researches from multiple disciplines at different scales together is novel because doing so is essential for further understanding land-atmosphere interaction. Many interesting results and insights are introduced across the manuscript, and I suppose the work will give a great contribution to land-atmosphere interaction research. Let me provide some comments that hopefully improve the manuscript.

Page 1 L20 (and related to photosynthesis parameters) I suppose the necessity of accurate leaf parameters to describe photosynthesis and stomatal conductance is such an obvious fact, and I am afraid if it can be any interesting finding. Instead, can the authors provide any insights with respect to how accurate leaf parameters are important for ABL? Leaf parameters greatly vary even among C3 plants (e.g., Miner et al., 2017), and it is quite common to see a global-scale terrestrial ecosystem model using different leaf parameters by different vegetation types (e.g., Sun et al., 2014). The original article by de Pury and Farquhar (1997) already manifests the importance of N effects on a photosynthetic capacity, and that may explain a large part of the differences in Am and alpha0 between the campaigns listed in Table 3. If the authors discuss how tuned leaf parameters affects ABL, I tempt to suggest looking at a parameter that determines stomatal conductance in the A-gs model in addition to two parameters already introduced. I suppose f0 is the one (P247 in Vilà-Guerau de Arellano et al., 2015), and it can be determined by the outputs of LI6400. For example, Ikawa et al., (2018) reports a leaf parameter for stomatal conductance (ml in their paper in Table 4) of rice considerably affects ABL temperature.

Answer: Following the advice of the referee we have extended and elaborated more on the impact of the optimized constants of the A-gs model using the CloudRoots data on the surface fluxes and the boundary layer height. We have added new information in section 3.6 to show that indeed the new constants used in the photosynthesis and stomatal resistance model presented at Table 3 influence and improve the model results compare to the observations. In particular, light-use efficiency ($\alpha$0). A refer-

ence is also included to show the relevance to connect leaf processes to surface and boundary-layer scale processes. To further support this we have included two new figures in the supplementary material to show the different on the evolution of fluxes and boundary-layer height between the default and the optimized CloudRoots constants.

L25 resolution of?

Corrected

L26 non-linear behavior to what?

Corrected L28 please spell out ET when introduced for the first time

Corrected

L29 inferred

Corrected

L2 evapotranspiration of total ET?

Corrected L9 Isn't "stomatal responses" a part of "surface and boundary-layer dynamics"?

We have rephrased the sentence

L17 Similarly, "wind" is a part of boundary-layer dynamics? It may be better to replace "BL dynamics" with other specific terms.

We have rephrased the sentence

L22 first is redundant

Corrected

L24 How large is a grid of weather and climate models?

We have included this information.

L10 diurnal variability of CO2-H2O flux partition

Corrected

L17 first is redundant

Corrected

L24 evapotranspiration or ET? Are they different?

Corrected

L5 mm instead of mm m-2, assuming 1 gram of H2O is 1 cubic centimeter. [0.1g]/[(0.2ˆ2*pi/4)m2] = [0.1cm3] /[(0.2ˆ2*pi/4)m2] = 0.1/ (0.2ˆ2*pi/4)/1000 mm

We have corrected the units.

L21 were

Corrected

L23 which leaf? Fully expanded? Or totally random in a canopy?

Corrected

L28 Please use the SI unit instead of ppm for CO2 concentration.

Corrected

L1 Parameterization based on a rapid A-Q curve depends on stomatal conductance. I advise to include an average gsw value during the measurement of PAR = 0 − 200

umol m-2s-1.

Answer: https://www.licor.com/env/support/LI-6800/topics/rapid-light-curve.html We have included these information in the paper

L14 superscript -2

Corrected

Page 8 Answer: Following the advice of the referee we have rewritten part of section 2.3.6 on the description of the scintillometer observations and the data analysis behind it,

L15 Considering the expertise of the authors, I will leave it to them whether to call it MO length or O length. http://glossary.ametsoc.org/wiki/Obukhov_length

We have corrected the name of the length scale: Obukhov length scale

L15-16 Please check the sentence.

We have rephrased the sentence L22 Please check the sentence.

We have rephrased the sentence

L11 2.105 may be too precise.

2.105 was incorrect. We have put the right value. It is 2.4 m.

L22 were

Corrected

L5 what surface? Plant-canopy?

Corrected

L19 Fig. 5

Corrected

L24 Table 3 and it is already mentioned in L20.

Corrected

L27 Without knowing local time (or solar time), it is difficult to catch up discussion.

We have added the times to facilitate the reading

L6 Fig. 3

Corrected

L10 How was latent heat flux from EC? Was it also high in IOP3 despite small gsw?

We have added additional information on the latent heat flux measurements observed during IOP 3 (still with large values of ET). The large decreased of ET occurred one week later.

L18 internal $CO_2$ concentration

Corrected

L25 Fig. 5

Corrected

L10 subscript 2

Corrected

L20-21 The presence (and not absence) of the local maxima increases thermal stability?

Corrected

L7 Please check the unit

Corrected

L16-17 Please note that detail profile does not necessarily provides accurate values of gross primary production at least given our limited knowledge of plant canopy micrometeorology, though the profile approach is still useful for understanding mechanisms (e.g., Drewry et al., 2014).

We have rewritten the sentence to be more precise and we have included the reference

L19 soil respiration or Rs?

Corrected

L26-27 Ep?

It has been clarified

L27 What is the sub-optimal performance?

We have added a sentence to explain better the sub-optimal performance. It is due to the assumptions in the partition method. Figures 7 and 9 have been also updated to correct some typos in the legends.

L25 It is a quite interesting topic that LE and $CO_2$ flux behave differently under diffuse and direct radiations. Can $CO_2$ flux also be included in Fig 10 to ensure the opposite pattern? It may be worth looking at a bulk stomatal conductance estimated from latent

heat flux to delineate the effects of conductance and VPD (e.g., Dolman et al., 1991).

Answer: Following the advice of the referee we have included an complementary figure to show the effect of clear and cloudy skies on the net primary production as a function of the photosynthetic active radiation during the same period. Section 3.3.1 has been modified accordingly.

L6 reaches

Corrected

L7 I prefer to know this information of time earlier.

We have placed the sentence before to facilitate the reading

I am afraid 3.4.2 needs a substantial improvement in the clarification. I was totally lost in the latter part of 3.4.2 and not able to understand how ET was estimated from SIF in Fig 16. I was not also able to understand what information authors want to convey by Fig 15. What is the correlation? It does not look like a correlation coefficient (0-1), and I do not see any relationships between x and y-axis either.

Answer: Following the advice of the referee we have rewritten the entire two sections to determine the variability of ET on space. We have divided the use of SIF in two sections: temporal and spatial variability. For the latter, we have rewritten the entire section 3.5 to explain better the method and the correlation between the estimations of ET and the SIF measurements. A more simple ne figure 15 has been made to facilitate the visualization of the relationships between ET and SIF.

L4 delete "it"

Corrected

L23 H or SH? Also some are italic and some are not.

We have been more consistent. It should be H.

L27 Can it be simply because that the greater the net heat received by both horizontal and vertical fluxes (surface flux and advection), the faster the ABL height grows?

Answer: Our explanation was referring to the sensible heat flux. The advection of warmer air leads to a reduction of H since the gradient between the skin temperature and the atmospheric temperature reduces.

L26 I apologize in advance for the lack of knowledge in the operational theory of scintillometer, but is the scintillometer measurement free from the concern of low frequency component? The spatial scale of 86.8 m over 60 secs by the Eurasian measurements may be still smaller than the low-frequency scales that would not be captured by EC measurements (greater than a spatial scale that would be inferred by the scale of tower height, wind speeds and time periods of 30 mins)?

Answer: As mentioned before section 2.3.6 has been rewritten to gain clarity. In short: The scintillometer method is based on structure parameters which are defined in the inertial range of the turbulent spectrum. Scintillometers determine the structure parameter looking at roughly one eddy size and average that effect in both time (relatively short compared to EC) and space (relatively long compared to EC). Essential to the method is that in contrast to EC it does not rely on resolving all eddy scales that contribute to the flux. The method does rely on MOST to link structure parameters to fluxes.

Answer: There was some text in the method section (2.3.6) on this:

Answer: "The added value of DBLS fluxes over the traditional EC method is that they

converge to statistically stable flux estimates at much shorter flux averaging times of 1 minute or less, while the EC technique typically requires flux averaging times of 10 to 30-minutes (Hartogensis et al, 2002; van Kesteren et al., 2013b)."

This has been extended with:

"The essence behind this is that the flux estimate is based on structure parameters which are defined in the inertial range of the turbulent spectrum. As such the flux estimates rely on a limited range of the turbulent scales that contribute to the flux rather than all as is the case with the EC method."

P24L16 those for the Corrected

L1 It sounds contradicting to the importance of using accurate leaf parameters. Maybe the variations of crop coefficients (K) are so small among different vegetation covers (0.7-1.1) compared to the difference between vegetative and non-vegetative (0 – 0.2) areas (Table 4)?

Answer: The section has been completely rechecked and rewritten. It now included a new classification of the land-use around CloudRoots.

Figs and Tables

Fig 4 Check the number of the fig.

Corrected

Fig. 7 I assume the dashed lines are canopy heights.

We have added this information in the figure caption

Fig. 16 Was this ET estimated by SIF? I still did not understand how it was estimated.

We have clarified this in the new section 3.5

Table 1 Height of wind measurements?

We have indicated in the caption of Table 1 that all the meteorological measurements were obtained at the height of $2.4 \pm 0.1$ m.

Table 2 Please check units.

Checked and corrected

Table 3 Which was determined by A-PAR and which was by A-Ci?

We have added this information

---

## Author Comment (AC2) · 9 Jul 2020

Response to reviewer RC2 Understanding sub grid variability of mass and energy fluxes, especially over heterogeneous landscapes or those subjected to full sun and shade under days with fair weather clouds remains a challenge to our field. This is an interesting effort to bring together observations and models to better understand these processes. I've always been curious about how sunlight can drive fluxes which in turn drive pbl growth and humidification of the boundary layer, which then affect humidity and the production of fair weather clouds. Hence a mix of positive and negative

feedbacks acting in concert.

1.- Answer: We thank Prof. Dennis Baldocchi for his evaluation. In particular his review shows the broad overview that helps to connects this CloudRoots paper with previous reserach. His suggestions to clarify some assumptions in treating the scintillometer data and the photosynthesis-conductance model are treated in this response that includes the modifications introduced in the revised manuscript.

This paper describes the detailed CloudRoots experiment. It joins a class of papers, like many of the key papers in the past that described FIFE, HAPEX, BOREAS, the Boardman ARM Field Experiment and various Kansas studies that provide the background for large investigator integrated field experiments. This newer generation of studies has some advantages over the past studies with the emergence of SIF as a proxy for photosynthesis. Plus there are better sensors for boundary layer height and more flux stations.

2.- Answer: We have included references to these previous campaigns to better connect our work to similar previous campaigns

Abstract/Introduction Overall it has the connections between leaf and soil, to canopy, to landscape, to region to boundary layer set of measurements and models to provide a rich database for discovery, model validation and model parameterization.

Methods

The team is using state of art eddy covariance measurements methods that are well vetted, though ICOS. Spatial integration is with scintillometers and inference of fluxes with remote sensing like SIF. The CLASS model is one of the best and is based on LES origins and can be used to drive a simpler one dimensional model. The scale of this work is over 10 km, which is reasonable. The study was in Germany and conducted as 3 intensive field experiments over the summer growing season. Land is flat and the landscape is well documented and assessed. Interesting to see mini lysimeters, soil

CO2 chambers and leaf physiological capacity measured, eg leaf gas exchange and sap flow, too. Well planned and executed.

My one complaint is use of MOST to interpret scintillometer measurements. They are often advocated to measure spatial averages, but the edges of the sampling will see advection, so there can be problems inverting fluxes with 2D measurements into a 1D framework. Eddy covariance works around this by establishing an internal boundary layer with large fetch.

3.- Answer: Unfortunately, until a better framework is developed to link statistical parameters (in this case $C_x^2$) to fluxes, we need to continue assuming Monin-Obukhov Similarity Theory (MOST). We agree that MOST should be used with care in conditions that violate the assumptions underlying MOST. An example of this is transition zones at field edges where measurements of $C_x^2$ are not only determined by the turbulent flux of the underlying surface but also from air advected from the neighboring field. During the CloudRoots experiment, we have checked that advection was not affecting our measurements for two reasons. First of all, the scintillometer transmitter and receiver are far enough from the edges of the field given the height of the sensor (1.95m), the wind speed and direction during the IOPs (see figure below), and the stability conditions. All of these make that footprints are small enough to fit within the field. A quick calculation for typical footprint length (90% footprint contribution) for the 3 IOPs yields: IOP1 (85m), IOP2 (30m) and IOP3 (75m). Second, the scintillometer has a path weighting function that is maximum in the middle of the path and near-zero at the transmitter and receiver positions. Therefore the main measurement of the signal occurs in the center of the field as figure 1 shows (see below). The figure also shows the main wind direction and speed during the three IOPs. The scintillometer application used in this paper (a laser scintillometer) should not be confused with the more common long-path scintillometers that are known for their capability to capture area-averaged fluxes at the landscape scale. Laser scintillometers can only be operated over short paths (<150m) and have the advantage that fluxes can be estimated at short time intervals, as is demonstrated in this paper. Another aspect of MOST is the ambiguity of the MOST functions. The choice of MOST functions mainly influences the size of the fluxes. As our focus is on the dynamical behavior of fluxes at short timescales in response to rapid changes in radiative forcing, the correct representation of the size of the fluxes is of secondary importance to us. In relation to the main manuscript, and at section 2.3.6, we have clarified the assumption of MOST and provide information on the footprints.

One thing I do like about scintillometer, and something this team has done, is look at instant fluxes with sun and shade and calculate changes in surface conductance. That approach to me has proven powerful and interesting. Van Kesteren, B., Hartogensis, O. K., Van Dinther, D., Moene, A. F., De Bruin, H. A. R., & Holtslag, A. A. M. (2013). Measuring H2O and CO2 fluxes at field scales with scintillometry: Part II–Validation and application of 1-min flux estimates. Agricultural and forest meteorology, 178, 88-105. All this information is then integrated through measurements of boundary layer devel- opment. Not sure of other such comprehensive field studies, that compete, even FIF, BOREAS or HAPEX.

4.- Answer: We agree with his comment that these one-minute fluxes are an original part of this CloudRoots reserach

Results Information on plant physiological performance is pertinent as it provides pa- rameter information for subroutines in CLASS. I'd like to know more about Vcmax, Jmax and the Ball Berry or Medlyn/Leuning type stomatal conductance parameters used by the model.

5.- Answer: We have included in the supplementary information a new Table S1 includ- ing the equivalence of the A-gs parameters to the values of Vcmax, Jmax and TPU at 298 K. The Table is located at the end of the response. . Regarding the formula- tion of the leaf and canopy conductances, we have provided an explanation in section 2.1 and the key references: Jacobs and De Bruin (1997), Ronda et al. (2001) and

[Figure]

Pedruzo-Bagazgoitia et al. (2017). The paper plays a new role to look at cloud induced fertilization of evaporation. Most studies focus on enhancement of co2 flux. So this is new and novel. Interesting to see At constant Q*, the median of LvE is always higher under clear skies than for cloudy skies. Diffuse fraction plays a minor role and the decrease on LvE under cloudy conditions is mainly due to the reduction in the incoming shortwave radiation. There remains some debate and discussion on the term evaporation over evapotranspiration. I favor the former after hearing John Monteith advocate for it. The paper drives towards a connection with large scale integrated SIF to landscape average evaporation. Since water and carbon fluxes are tightly coupled, I am open minded to this spatial scaling assessment. At boundary-layer integrated scale, they find that modelled sensible heat flux correlates better with the area weighted average flux than the local flux estimates. I find this interesting as we are dealing with a similar problem. How to best average fluxes from a network of flux towers as a lower boundary condition for a pbl model? Or do we get integration of fluxes at 30 m scale with ECOSTRESS?

5.- Answer: In calculating the sensible heat flux as the area-weighted flux we were limited by the eddy-covariance spatial measurements. Therefore this aggregated sensible heat flux is only based on two values taken above two different land conditions as shown in Figure 1. We appreciated very much the suggestion of the 30 meter data of ECOSTRESS, and we plan to use it in future CloudRoots studies that aim to investigate the impact of surface heterogeneity on the boundary-layer and cloud-dynamics. In this paper, we have also included an estimation of the heterogeneity of evapotranspiration inferred from the SIF data (see Figure 14). As shown in Figure 18c, the aggregated of this estimation is slightly higher than the more local measurements of evapotranspiration.

This leads me to a suggestion. The PI should also look at ECOSTRESS data for their domain and compare the integrated evaporative fluxes with what they are producing. These data are publicly available and could be a nice alternative constraint. Fisher, J.

B., et al. (2020), ECOSTRESS: NASA's Next Generation Mission to Measure Evapo-transpiration From the International Space Station, Water Resources Research, 56(4), e2019WR026058, doi:10.1029/2019wr026058.

[Figure]

Figure 1: Scintillometer transmitter and receiver locations in relation to the dimensions of the CloudRoots field. The blue arrows indicate the wind-direction on the 3 IOPs with the wind-speed at z=2m.

5.- *Answer*: We have included in the supplementary information a new Table S1 including the equivalence of the A-gs parameters to the values of Vcmax, Jmax and TPU at 298 K. The Table is:

**Table S1**: Equivalent temperature-normalized maximum carboxylation, electron transport and triose phosphate utilization rates ($V_{cmax25}$, $J_{max25}$ $TPU_{25}$, respectively) commonly used in the Farquhar-Berry-von Caemmerer (FBvC) model of leaf photosynthesis (Farquhar *et al.*, 1980). Fits between the FBvC and A-$g_s$ models were obtained using the plantecophys package (Duursma, 2015) based on A-$g_s$ model output with parameter values noted in Table 3, a constant temperature of 25°C and PAR of 1500 $\mu$mol m$^{-2}$ s$^{-1}$.

| Parameter setting (table 3) | $V_{cmax25}$ ($\mu$mol m$^{-2}$ s$^{-1}$) | $J_{max25}$ ($\mu$mol m$^{-2}$ s$^{-1}$) | TPU ($\mu$mol m$^{-2}$ s$^{-1}$) | RMSE |
|---|---|---|---|---|
| Default | 81.11 | 240.95 | - | 7.53 |
| IOP1 | 98.37 | 196.43 | 12.97 | 5.96 |
| IOP2 | 93.69 | 121.09 | 7.58 | 2.16 |
| IOP3 | 15.43 | 15.77 | 0.82 | 2.56 |

**Fig. 1.**